# Learning 3D-Gaussian Simulators from RGB Videos

**Mikel Zhobro** [1]   **A. René Geist** [† 1]   **Georg Martius** [† 1]

## Abstract

Realistic simulation is critical for applications ranging from robotics to animation. Video generation models have emerged as a way to capture real-world physics from data, but they often face challenges in maintaining spatial consistency and object permanence, relying on memory mechanisms to compensate. As a complementary direction, we present 3DGSim, a learned 3D simulator that directly learns physical interactions from multi-view RGB videos. 3DGSim adopts MVSplat to learn a latent particle-based representation of 3D scenes, a Point Transformer for the particle dynamics, a Temporal Merging module for consistent temporal aggregation, and Gaussian Splatting to produce novel view renderings. By jointly training inverse rendering and dynamics forecasting, 3DGSim embeds physical properties into point-wise latent features. This enables the model to capture diverse behaviors, from rigid and elastic to cloth-like dynamics and boundary conditions (e.g., fixed cloth corners), while producing realistic lighting effects. We show that 3DGSim can generate physically plausible results even in out-of-distribution cases, e.g. ground removal or multi-object interactions, despite being trained only on single-body collisions. For more information, visit: `https://mikel-zhobro.github.io/3dgsim`.

## 1. Introduction

Simulating visually and physically realistic environments is a cornerstone for embodied intelligence. Robots must soon tackle tasks such as opening washing machines, folding laundry, or tending plants. Traditional analytical simulators demand exact geometry, poses, and material parameters,

† Equal advising. [1]University of Tübingen, Germany. Correspondence to: Mikel Zhobro <mikel.zhobro@uni-tuebingen.de>.

*Proceedings of the $43^{rd}$ International Conference on Machine Learning*, Seoul, South Korea. PMLR 306, 2026. Copyright 2026 by the author(s).

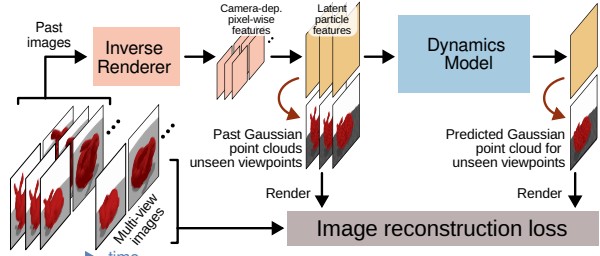

*Figure 1.* **3DGSim Overview.** An inverse renderer extracts **latent features** from multi-view RGB history. These are evolved by a **transformer-based dynamics model** and decoded into **3D Gaussian Splats** for **novel-view next-frame reconstruction**.

making arbitrary scene simulation impractical. An alternative is to learn models that predict future states of a scene in large-scale observations, as evidenced by the striking visual realism of 2D video generation methods (Li et al., 2022; Wu et al., 2023; NVIDIA et al., 2025). However, pure 2D approaches lack 3D structure awareness, leading to failures in occlusion handling, object permanence, and physical plausibility (Motamed et al., 2025).

3D-based representations address many of these shortcomings, as shown by recently learned particle-based simulators (Li et al., 2019; Allen et al., 2023; Zhu et al., 2024) which model a wide range of physical phenomena, from fluids and soft materials to articulated and rigid body dynamics. Yet, scaling such methods to data-rich regimes remains challenging, as most methods require privileged signals (object-level tracks, depth sensors, physics prior) or hand-crafted graph constructions.

To bridge this gap, we identify three pillars for generalizable, scalable visuo-physical simulation from videos: *(1) 3D visuo-physical reconstruction* from raw RGB observations; *(2) Imposing minimal physical biases* that can capture diverse physics; *(3) Efficient, differentiable decoding* back to image space for supervision via reconstruction loss.

Graph neural networks (GNNs) (Sanchez-Gonzalez et al., 2020; Xue et al., 2023; Whitney et al., 2023; 2024; Shi et al., 2024; Wang et al., 2024a) have shown great promise in introducing relational inductive biases to handle the unstructured nature of particle sets. This has allowed GNN-based particle simulators to make major progress on all three pillars. In particular, (Whitney et al., 2023) jointly train an encoder

and dynamics model to learn visuo-physical pixel features from RGBD, and in the follow-up work (Whitney et al., 2024) eliminate point correspondences via abstract temporal nodes or per-step models with merging. (Driess et al., 2023) demonstrate end-to-end dynamics training of composable NeRF fields from raw RGB images. These advances, in combination with recent advances in feed-forward inverse rendering (Chen et al., 2024) and fast differentiable rendering of particles (Kerbl et al., 2023), encourage us to ask the question: *can we give up the inductive bias arising from locally connected graphs and still learn 3D particle-based simulators?*

To this end, we build 3DGSim, a fully end-to-end differentiable framework that embraces the power of scalable computation over hand-crafted biases. 3DGSim begins by inferring 3D visuo-physical features from raw multi-view RGB images through a feed-forward inverse renderer based on MVSplat. We then introduce a transformer-only dynamics engine, avoiding kNN-based graph construction and manually designed edge features in favor of learned spatiotemporal embeddings. Finally, a Gaussian Splatting head enables training on an image reconstruction loss from multi-view videos.

Specifically, 3DGSim makes the following contributions:

- **Inverse Renderer**: Extends MVSplat with a feature extraction module fusing pixel-aligned features into a particle visuo-physical latent representation.

- **Temporal Encoding & Merging Layer**: Discards abstract temporal nodes in favor of a hierarchical module that processes an arbitrary number of timesteps.

- **Transformer-Only Dynamics Engine**: Removes graph biases and instead uses space-filling curves and learned embeddings for particle-based simulation.

- **End-to-End Differentiable Framework**: Connects inverse rendering, transformer dynamics, and Gaussian splatting-based decoding to learn next-frame image reconstruction.

- **Open Source Release**: We release the code and dataset to establish a reproducible baseline for future visuo-physical simulation research.

## 2. Related work

**Encoding and rendering scene representations.** Common 3D scene representations include point clouds (particles), meshes, signed distance functions (SDFs), neural radiance fields (NeRFs) (Mildenhall et al., 2021), and 3D Gaussians (splats) (Kerbl et al., 2023). Point clouds, which approximate object surfaces, can be obtained from RGB-D sensors (Shi et al., 2024; Whitney et al., 2023; 2024) or via inverse rendering (Wang et al., 2025; Murai et al.,

2024; Chen et al., 2024). Works, such as (Whitney et al., 2023; 2024), use U-Net–style encoders trained jointly with the dynamics model, allowing the extracted features to be optimized for physical prediction, a strategy shown to outperform independently trained encoders (Li et al., 2022). We adopt this joint training approach using MVSplat (Chen et al., 2024), where the encoded features are initially bound to camera parameters. To unify these visuo-physical latents in a global frame, we introduce a learned feature transformation module that maps them into a consistent 3D representation. While many PBS methods render from NeRFs (Xue et al., 2023; Shi et al., 2024; Whitney et al., 2023; 2024; Wang et al., 2024a; Driess et al., 2023), we instead encode visual appearance directly in the particle cloud using 3D Gaussians. This explicit representation offers high rendering fidelity and significantly improved efficiency over NeRF-based rendering (Kerbl et al., 2023), supporting scalability.

**GNN based particle-based simulators (PBS).** Graph neural networks (GNNs) introduce relational inductive biases well-suited for modeling the unstructured nature of particle systems. Early work (Sanchez-Gonzalez et al., 2020; Li et al., 2019) demonstrated that GNN-based PBS can fit trajectories across a range of physical phenomena. However, GNNs struggle with rigid bodies, where instantaneous velocity changes require long-range message passing across the entire graph in a single step. To address this, later works incorporate mesh structures (Pfaff et al., 2021; Allen et al., 2022) or signed distance functions (SDFs) (Rubanova et al., 2024) to enforce object-level coherence. Although effective in rigid-body settings, these methods do not generalize to deformable or fluid systems. Recent works (Saleh et al., 2024; Whitney et al., 2024) suggest adding attention layers to efficiently pass information through the graph. (Wang et al., 2024a) move toward greater data efficiency by incorporating physics-inspired biases such as the Material Point Method, though limiting broad applicability and requiring small simulation timesteps. To address temporal correspondence, (Whitney et al., 2023) introduces abstract temporal nodes, while (Whitney et al., 2024) combines GNNs with transformers to improve memory efficiency by processing and merging pairs of timesteps. However, the method is restricted to two-step horizons, as it requires training a separate model for each additional timestep. Methods based on GNNs rely on kNN to define point connectivities within a fixed radius and hand-crafted features based on object associations and distances to define graph features. Message passing and spatial pooling via furthest-point-sampling (FPS) are then used to aggregate information for dynamics prediction. However, kNN and distance computations are expensive and take up 54% of the forward time (Wu et al., 2024b), which limits scalability and prevents real-time forecasting. In contrast, we follow the design of PTv3

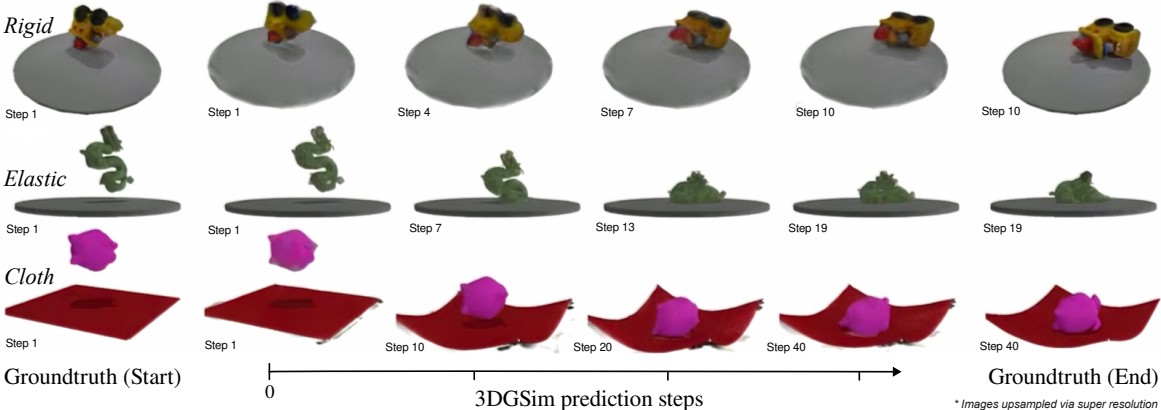

*Rigid*

Step 1 — Step 1 — Step 4 — Step 7 — Step 10 — Step 10

*Elastic*

Step 1 — Step 1 — Step 7 — Step 13 — Step 19 — Step 19

*Cloth*

Step 1 — Step 1 — Step 10 — Step 20 — Step 40 — Step 40

Groundtruth (Start) — 0 — 3DGSim prediction steps — Groundtruth (End)

*\* Images upsampled via super resolution*

*Figure 2.* Qualitative examples of 3DGSim's dynamic predictions. After training on less than 6 minutes of video per object across 6 objects, 3DGSim accurately predicts the motion of elasto-plastic deformations, rigid bodies, and cloth. The first and last column represent the initial and last frames of the "true" simulated motion. The in-between columns/frames are predictions of 3DGSim.

(Wu et al., 2024b). In 3DGSim, we trade off exact KNN neighborhood computation with space-filling curve–based ordering of particles and use sparse convolutions to encode relative positions, avoiding distance calculations. To enable the processing of temporal point clouds, we propose Temporal Merging with Grid Pooling to construct a hierarchical spatiotemporal UNet-style Point Transformer for dynamics prediction.

**Analytical particle simulators as physical prior.** Our work differs in purpose from applications which use Gaussian Splatting particles and analytical PBS as physical prior (e.g. off-the-shelf differentiable MPM simulator) to accomplish a series of tasks such as tracking (Luiten et al., 2024; Keetha et al., 2024; Zhang et al., 2024a; Abou-Chakra et al., 2024), dynamic scene reconstruction (Wu et al., 2024a; Huang et al., 2023; Yu et al., 2023)), or animation (Xie et al., 2023; Zhang et al., 2024b; Lin et al., 2025b). While analytical PBS can be used for parameter identification (Abou-Chakra et al., 2024), they are tailored to specific simulation scenarios. For a detailed comparison, refer to the supplementary material (see Appendix D.2).

## 3. Preliminaries

3DGSim is built atop several prior works, namely: 3D-Gaussian splatting, which enables fast rendering, MVSplat, which yields 3D Gaussian point clouds from multi-view images, and PTv3 which enables efficient neural processing of 3D point clouds.

**Gaussian Splatting.** 3D Gaussian splatting (3DGS) (Kerbl et al., 2023) is an effective framework for multi-view 3D image reconstruction, representation, and fast image rendering and has gained rapid popularity due to its support for rapid inference, high fidelity, and editability of scenes.

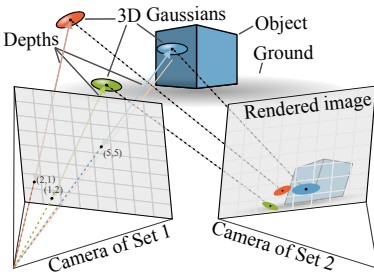

*Figure 3.* MVSplat uses a cost volume with plane sweeping to regress pixel-wise 3D Gaussian parameters.

Gaussian splatting uses a collection of 3D Gaussian primitives, each parameterized by

$$g_i = (p_i, c_i, r_i, s_i, \sigma_i), \tag{1}$$

with the Gaussian's mean $p_i$ (particle position), its rotation $r_i$, spherical harmonics $c_i$ (defines coloring), scale $s_i$, and opacity $\sigma_i$. To render novel views, these primitives are projected onto a 2D image plane using differential tile-based rasterization. The color value at pixel $\mathbf{p}$ is calculated via alpha-blend rendering: $I(\mathbf{p}) = \sum_{i=1}^{N} \alpha_i c_i \prod_{j=1}^{i-1}(1 - \alpha_j)$ where $\alpha_i = \sigma_i e^{-\frac{1}{2}(\mathbf{p}-p_i)^\top \Sigma_i^{-1}(\mathbf{p}-p_i)}$ is the 2D density, $I$ is the rendered image, $N$ is the number of primitives in the image and $\Sigma_i$ is the covariance matrix given by $\Sigma_i = r_i s_i r_i^\top$ for improved computational stability.

**MVSplat: Multi-view feed-forward 3D reconstruction.** MVSplat deploys a feed-forward network $f_\phi$ with parameters $\phi$ that maps $M$ images $\mathcal{I} = \{I^m\}_{i=m}^{M}$ with $I^m \in \mathbb{R}^{(H \times W \times 3)}$ to a set of pixel-aligned 3D Gaussian primitives (Fig. 3)

$$f_\phi : \{(I^m, P^m)\}_{m=1}^{M} \mapsto \{g_i\}_{i=1}^{M \times H \times W}. \tag{2}$$

At each time step, MVSplat localizes Gaussian centers using a cost volume representation through plane-sweeping and

cross-view feature similarities. To do so, it requires the corresponding camera projection matrices $\mathcal{P} = \{P^m\}_{m=1}^M$ that are calculated as $P^m = K^m[R^m|t^m]$ via the camera intrinsics $K^m$, rotation $R^m$, and translation $t^m$.

# 4. 3DGSim

3DGSim is a fully differentiable pipeline that, given $T$ past multi-view RGB frames, reconstructs 3D particles with latent features, simulates their motion, and renders the next frames. It consists of three jointly trained modules (Fig.1): (i) an *encoder* that maps multi-view RGB images to 3D particles, (ii) a *dynamics model* that simulates the motion of these particles through time, and (iii) a *renderer* that yields images by first mapping the particles to Gaussian splats.

## 4.1. State Representation

To simulate physical scenes from vision, we require a state representation that is both expressive enough to capture fine-grained 3D and physical properties, and compact enough to enable efficient learning and prediction. Although an explicit 3DGS representation $g_i(t_k)$ offers geometric and visual completeness, it is insufficient for dynamics modeling. Instead, we distill the state of each particle into a more compact representation:

$$\tilde{g}_i(t_k) = \big(p_i(t_k),\, f_i(t_k)\big) \tag{3}$$

where $t_k$ denotes the $k$-th timestep, $p_i$ the position, and $f_i \in \mathbb{R}^d$ the visuo-physical latent particle feature, encoding shape, appearance, and dynamic properties. Unless otherwise stated, we omit the timestep $t_k$ and the particle index $i$ when the statement applies to all timesteps or particles, respectively.

**Optional: Masking and Freezing of Particles.** At each timestep $t_k$, the encoder yields pixel-aligned features for each input image. As an optional step, one can apply a foreground mask to discard particles likely belonging to the static background, retaining a reduced set of $N_k$ particles per time step (Fig. 3). Additionally, as originally suggested by (Whitney et al., 2023), static particles can optionally be "frozen", i.e. act as input to the dynamics model but are excluded from position updates. These optional strategies improve efficiency without being necessary for successful training, as shown in Section 5 and Appendix C.

**Invariant and dynamic feature decomposition.** We decompose each particle's visuo-physical feature into an invariant and a dynamic part as shown in Fig. 4, writing

$$f_i = f_i^{\text{inv}} \oplus f_i^{\text{dyn}}, \tag{4}$$

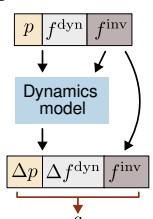

*Figure 4.* $p$ and $f^{\text{dyn}}$ are updated, $f^{\text{inv}}$ stays constant.

where $\oplus$ denotes concatenation. The dynamics model updates only $f_i^{\text{dyn}}$, while leaving $f_i^{\text{inv}}$ unchanged. For clarity, we will refer to the dynamics update as "updating $f_i$", though only the dynamic component $f_i^{\text{dyn}}$ is altered.

## 4.2. View-Independent Inverse Renderer

In MVSplat, pixel-aligned features $\hat{f}_i'$ are tied to the specific camera view from which they were extracted. While Gaussian primitives (e.g. depth, scale, rotation, harmonics) can be directly unprojected or transformed into the world frame using camera parameters, latent features remain bound to the camera-centric frame. Since dynamics predictions are invariant to the observer's viewpoint, such a dependence on view-dependent encodings hampers generalization.

To overcome this, 3DGSim introduces a feature encoding network that maps pixel-aligned features $\hat{f}_i'$ into view-independent latent representations $f_i$. The encoder employs FiLM conditioning (Perez et al., 2017) on pixel depth, pixel shift, density, and ray geometry (parameterized via Plücker coordinates (Plücker, 1868-1869)) to infer spatially consistent 3D features. As a result, the inverse rendering module produces canonically anchored particle states, providing a unified representation for downstream dynamics learning. Further architectural details are described in Appendix A.1.

## 4.3. Dynamics Model

At the core of our method is the dynamics model, a transformer architecture operating on particle sets in space and time. The dynamics model receives as input $T$ past particle sets, $\big\{\{\tilde{g}_i(t_k)\}_{i=1}^{N_k}\big\}_{k=1}^T$, where $\tilde{g}_i(t_k) = \big(p_i(t_k),\, f_i^{\text{inv}}(t_k),\, f_i^{\text{dyn}}(t_k)\big)$, and predicts the updated dynamic features at the next timestep

$$\Delta p(t_T), \Delta f^{\text{dyn}}(t_T) = \text{Dyn Model}\left(\big\{\{\tilde{g}_i(t_k)\}_{i=1}^{N_k}\big\}_{k=1}^T\right), \tag{5}$$

such that $p_i(t_{T+1}) = p_i(t_T) + \Delta p_i(t_T)$ and $f_i^{\text{dyn}}(t_{T+1}) = f_i^{\text{dyn}}(t_T) + \Delta f_i^{\text{dyn}}(t_T)$. As these point clouds are unstructured and potentially vary in size at each time step due to masking, a fundamental challenge arises: *How can a network efficiently propagate the embedded physics information both spatially and temporally?*

We tackle this question by building on PTv3 (Wu et al., 2024b), which has recently achieved state-of-the-art performance in representation learning for unstructured point clouds (Wu et al., 2025). As discussed in Appendix A.2, PTv3 operates by serializing the input point cloud and applying patch-wise attention. However, the original design of PTv3 is limited to point clouds that do not exhibit temporal variation. In this section, we extend PTv3 to predict dynamics from *temporally evolving point clouds*. First, we extend serialization to equip point cloud encodings with a times-

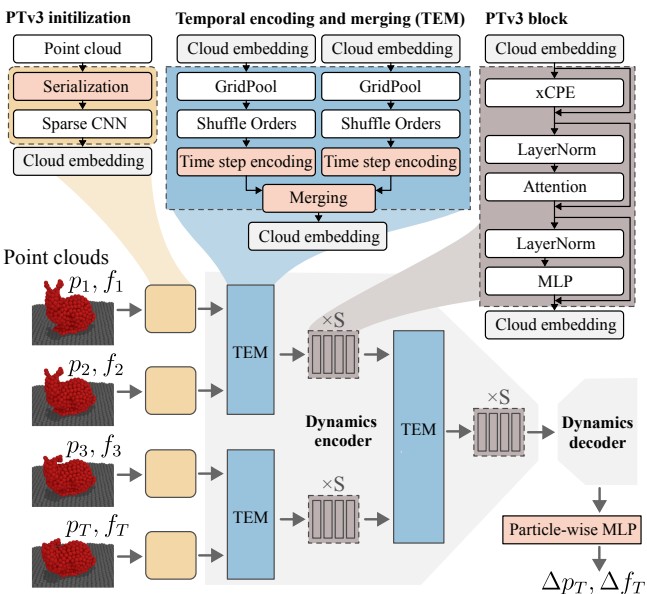

*Figure 5.* **The dynamics model** encodes the time step into each embedding and merges embeddings from adjacent timesteps. The TEM and PTv3 blocks are applied repeatedly until all embeddings are merged. Our extensions to PTv3 are highlighted in red.

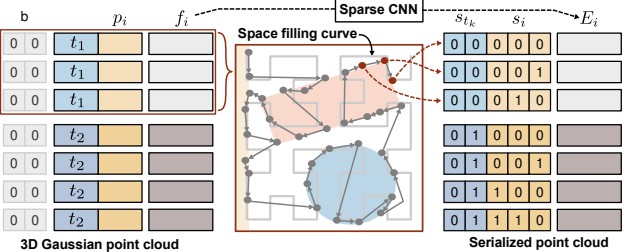

*Figure 6.* Spatio-temporal point cloud serialization.

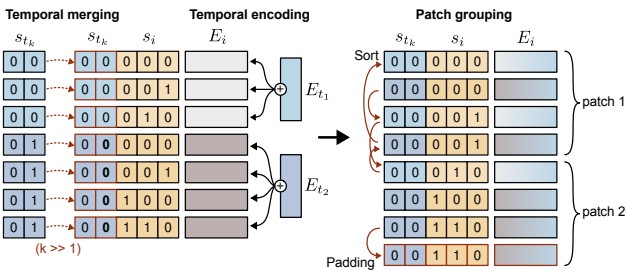

*Figure 7.* Temporal merging and embedding followed by patch grouping for applying patch-wise attention.

tamp. Then, we equip features with temporal embeddings that allow attention to distinguish timestamps. Lastly, we use the timestamps to merge neighboring latent particle sets, enabling PTv3's patch-wise attention blocks to aggregate information across time.

**Temporally serialized point cloud (t-SPC).** To enable spatio-temporal reasoning over multiple timesteps, we ex-

tend PTv3's point serialization scheme by encoding both spatial and temporal structure into a single key. Specifically, for each particle $i$ at timestep $t_k$ in batch $b$, we define a 64-bit serialization code:

$$\tilde{s}_i(t_k, b) = \Big[ \underbrace{b}_{(64 - \tau - \kappa)\,\text{Bits}} \mid \underbrace{s_{t_k}}_{\tau\,\text{Bits}} \mid \underbrace{s_i}_{\kappa\,\text{Bits}} \Big] \quad (6)$$

Here, $s_{t_k}$ is the temporal code and $s_i$ is a spatial code obtained by projecting $p_i$ onto a space-filling curve (SFC). We set $\kappa = 48$ and allocate $\tau = \log_2(T)$ bits for time. With 16 bits per dimension and a grid resolution of $G = 0.004\,\text{m}$, the spatial encoding spans up to $216\,\text{m}$ per axis.

**Temporal encoding.** As shown in (Fig. 7), before merging t-SPCs across timesteps, we inject a learned, timestep-specific positional encoding $E_{t_k}$ as

$$f_i(t_k) \leftarrow f_i(t_k) + E_{t_k}. \quad (7)$$

This temporal encoding ensures that the attention mechanism can distinguish points across different temporal instances, enabling the model to reason about dynamics over time. Similar positional encoding methods have previously been applied in transformer architectures to differentiate positions within sequences (Vaswani et al., 2017).

**Temporal merging.** Unlike PTv3, which restricts attention exclusively to patches composed of points from the *same time step*, our method enables a wider receptive field across time. To do so, we propose *temporal merging* which applies a one-bit right shift to the temporal codes $s_{t_k}$:

$$\text{Merge}(\tilde{s}_i) = [b \mid (s_{t_k} \gg 1) \mid s_i]. \quad (8)$$

For instance, points from time steps $s_{t_1} = 0$ and $s_{t_2} = 1$ are merged by shifting their codes, so they both become 0, as depicted in (Fig. 7). By grouping points from separate time steps into a single patch, the attention module can model relationships across time.

Importantly, while (Whitney et al., 2024) deploy a dedicated transformer module for each time step, our proposition of temporal merging enables the reuse of the same attention block across time steps, which significantly reduces memory consumption and promotes knowledge transfer.

**Patch-wise attention and particle-wise MLP.** After each temporal encoding and merging (TEM) block, the cloud embeddings are processed by PTv3's patch-wise attention block. First, the embeddings are equipped with a position encoding via a sparseCNN with skip connection (xCPE in Fig. 5). Then, the embeddings are fed to a patch-wise attention layer. Finally, at the end of the dynamics model, each particle alongside its embedding is mapped by a particle-wise MLP to $\Delta p_T$ and $\Delta f_T$.

## 4.4. Reconstruction Loss

To render images with 3DGS, particle states $\tilde{g}_i = (p_i, f_i)$ are transformed into Gaussian splat parameters $g_i$ via a learned head, materialized only at the final stage to supervise the training with image reconstruction.

3DGSim is trained using a single image reconstruction objective $\mathcal{L}$, computed from rasterized multi-view images. These images are rendered from both encoder-predicted past point clouds $\{\{g_i(t_k)\}_{i=1}^{N_k}\}_{k=0}^{T}$ and the simulated future trajectory $\{\{g_i(t_k)\}_{i=1}^{N_k}\}_{k=T+1}^{T+T'}$. The overall loss is defined as

$$\mathcal{L} = (1-\lambda)\frac{1}{T}\sum_{k=0}^{T}\mathcal{L}_k + \lambda \sum_{k=T+1}^{T+T'} \gamma^{k-T-1}\,\mathcal{L}_k, \qquad (9)$$

where the per-frame image loss $\mathcal{L}_k$ is given by

$$\mathcal{L}_k = \mathcal{L}_2(I_k^{\text{gt}}, I_k) + \beta\,\mathcal{L}_{\text{LPIPS}}(I_k^{\text{gt}}, I_k). \qquad (10)$$

We set $\lambda = 0.5$, the temporal decay factor $\gamma = 0.87$, $T \in \{2, 4\}$, $T' = 12$, and $\beta = 0.05$.

## 5. Experiments

In what follows, we train 3DGSim on different datasets and test the model's ability to generalize.

**Model setup.** Unless stated otherwise, the following training and parameter settings serve as defaults in the experiments. The state consists of dynamic $f^{\text{dyn}}$ and invariant features $f^{\text{inv}}$ of size $(32, 32)$ for the implicit- and $(n_f, 16)$ for the explicit 3D Gaussian particle representation. In the explicit representation, $f^{\text{dyn}}$ corresponds to explicit Gaussian primitives of size $n_f$ which are directly used for rendering. The inverse rendering encoder follows MVS-plat, reducing candidate depths from 128 to 64 due to smaller scene distances. Default near-far depth ranges are $[0.2, 4]$ for rigid bodies and $[1.5, 8]$ for the other datasets, as the scene has a larger scale. The dynamics transformer defaults to PTv3 with a 5-stage encoder (block depths $[2, 2, 2, 6, 2]$) and a 4-stage decoder ($[2, 2, 2, 2]$). Grid pooling and temporal merging strides default to $[1, 4, 2, 2, 2]$ and $[1, 2, 2, 2, 2]$, respectively, with grid size $G = 0.004$ m. Attention blocks use patches of size 1024, encoder feature dimensions $[32, 64, 128, 256, 512]$, decoder dimensions $[64, 128, 256]$, encoder heads $[2, 4, 8, 16, 32]$, and decoder heads $[4, 4, 8, 16]$. For the camera setup, we select 4 uniformly distributed views at random and an additional 5 target cameras from the remaining cameras (out of 12 total) to compute the reconstruction loss.

**Training.** Our models are trained with AdamW for $\sim$120,000 steps using a cosine annealing warm-up and a learning rate of $2 \times 10^{-4}$, with batch sizes of 6 and 4 for 2-step and 4-step states, respectively. To optimize memory and speed, we use gradient checkpointing and flash attention v2 (Dao, 2024). Training is performed on a single H100 GPU and typically takes around six days.

**Datasets.** To evaluate 3DGSim's robustness in learning dynamics from videos, we introduce three challenging datasets: rigid body, elastic, and cloth. The rigid body dataset consists of 1,000 simulated trajectories from the MOVI dataset, involving six rigid objects (turtle, sonny school bus, squirrel, basket, lacing sheep, and turboprop airplane) from the GSO dataset (Downs et al., 2022). Each trajectory spans 32 frames at 12 FPS, providing controlled dynamics characteristic of rigid body motion. The elastic dataset, aimed at capturing plastic deformable object dynamics, includes six objects (dragon, duck, kawaii demon, pig, spot, and worm) simulated using the Genesis MPM elastoplastic simulator (Authors, 2024). Each object undergoes deformation upon collision with a circular gray ground, offering scenarios of complex elastic behavior. The cloth dataset includes the same set of objects as the elastic dataset. Here, the cloth is fixed at four corners, posing the challenge to infer implicit constraints and model dynamic cloth-like deformations. Both elastic and cloth datasets include 200 trajectories per object, simulated with a 0.001 time step and 20 substeps. Each two second sequence is recorded at 42 FPS resulting in 84 frames per trajectory and less than 6 minutes of footage per object.

**Benchmarking.** Most existing 3D baselines do not permit direct comparison without substantial reimplementation. In particular, VPD, HD-VPD, DEL, and 3D-IntPhys (Whitney et al., 2023; 2024; Wang et al., 2024a; Xue et al., 2023) neither release code nor datasets, including upon author request, making reproducible evaluation infeasible. DPI-Net and VGPLDP (Li et al., 2019; 2020) are open-source, but require ground-truth particle trajectories and significant adaptation to our setting.

For 2D baselines, we compare against Cosmos (NVIDIA et al., 2025), which is pretrained on 5 past frames from a single view. We evaluate both the base and a LoRA-finetuned Cosmos-Predict2 variant (CosmosFT) trained for 6,000 iterations on our dataset using the recommended parameters. Results are presented in (Table S1, Appendix B) .

For 3D baselines, we compare against Spring-Gauss (Zhong et al., 2024), which samples anchors from a 3DGS reconstruction, connects them with springs, and uses a differentiable simulator backbone (details in Appendix B.1).

For evaluation, we randomly hold out 12% of the trajectories from each dataset. We report each model's PSNR, SSIM, LPIPS, and FVD in (Appendix B).

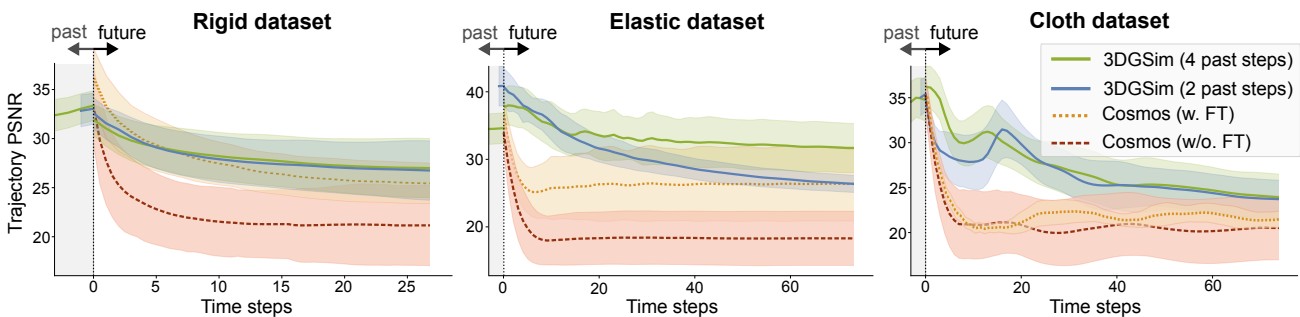

*Figure 8.* Trajectory PSNR of 3DGSim, Cosmos, and CosmosFT is shown for both past and future predictions. The Cosmos models are conditioned on past frames and appropriate language prompts.

## 5.1. Trajectory Simulation

3DGSim achieves competitive long-horizon simulation accuracy; up to 80 steps; compared with state-of-the-art baselines such as Cosmos-1.0-Autoregressive-5B-Video2World and a LoRA-finetuned Cosmos-Predict2 (NVIDIA et al., 2025). Performance curves are shown in (Fig. 8). Ablation studies (Tab. 1) reveal that keeping 3DGS primitives explicit in the representation yields similar short-term performance but generalizes poorly, especially with fewer cameras (see Appendix C). By contrast, using a latent implicit representation leads to more robust generalization.

## 5.2. Ablation

*Table 1.* Ablation Study: The table reports the performance of the Final Model (3DGSim) alongside variants where individual elements, such as Temporal Encoding/Merging, rollout length, feature representation, modality, and input steps, are removed or modified. The notation *4-1-12* indicates: a 4-step input window, predicting deltas only for the last step, and training with a 12-step future rollout.

| Variant | PSNR↑ | Δ |
|---|---|---|
| **Final (Latent 4-1-12)** | **33.15 ± 3.51** | **0.00** |
| Input Steps (2-1-12) | 32.05 ± 3.48 | -1.10 |
| Modality (Predict All 4-4-12) | 31.98 ± 3.78 | -1.17 |
| Explicit 3DGS (4-1-12) | 29.69 ± 1.75 | -3.46 |
| Rollout Length (4-1-2) | 26.43 ± 3.48 | -6.72 |
| Without TEM (4-1-12) | 18.19 ± 1.29 | -14.96 |

We use the elastic dataset to ablate the contribution of all key components of 3DGSim. **TEM is Critical:** Removing the Temporal Encoding and Merging (TEM) module causes a massive performance collapse (dropping to ~18 PSNR), indicating that simple spatial attention is insufficient.

**Latent > Explicit:** The latent representation (Final Model) outperforms the explicit 3DGS representation (29.69 PSNR). By keeping the representation abstract, the model can embed physical properties (like mass or friction) into the latent space instead of overfitting to the visual aspects.

**Rollout Length:** Training on longer horizons (12 steps) sig-

nificantly boosts performance compared to short horizons (2 steps, 26.43 PSNR), likely due to better long-term stability learning.

**Input Window**: A 4-step window outperforms the 2-step variant (+1.10 dB), confirming that a longer temporal context is required to accurately infer latent dynamics like velocity and acceleration. We also evaluate variable-length input windows at inference time on the elastic dataset and present results in (Appendix B.2).

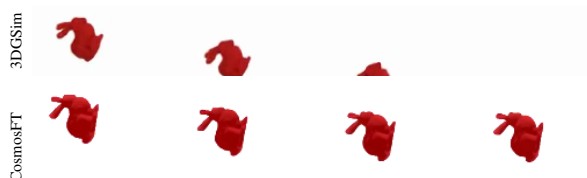

*Figure 9.* When the ground is removed, 3DGSim predicts the freefall, while CosmosFT hallucinates a levitating object at ground level.

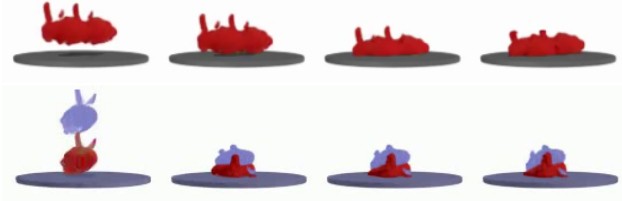

*Figure 10.* Although not trained on this specific elastic object or multiple objects, 3DGSim predicts physically plausible deformations.

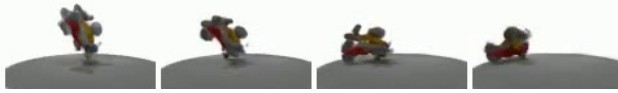

*Figure 11.* 3DGSim's prediction of a rigid plane captures shadows by altering ground particle appearance.

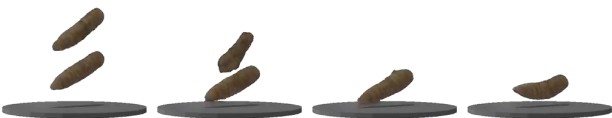

*Figure 12.* CosmosFT merges distinct worms into one before ground contact, even on in-distribution cases.

## 5.3. Scene Editing and Model Generalization

With its explicit 3D state, 3DGSim supports direct scene editing, providing a natural testbed for generalization. When the ground is raised or removed, conditions never seen during training, the model continues to generate stable, physically consistent rollouts (Fig. 9). This suggests a robust grasp of underlying dynamics that extends beyond the training distribution.

We further test generalization by duplicating objects and running long-horizon simulations (Fig. 10, Appendix F.1). Although trained only on single-object–ground collisions, 3DGSim accurately captures realistic multi-body interactions, with objects retaining integrity rather than collapsing into chaotic overlaps. Beyond interactions, it even models emergent properties such as shadows (Fig. 11), indicating a holistic understanding of lighting, geometry, and physics.

In contrast, CosmosFT struggles under similar 2D-edits. When the ground is removed, objects often remain suspended (Fig. 9), and when multiple objects are introduced, they morph into a single mass before contact (Fig. 12). These hallucinations reflect the limits of 2D image-based reasoning, underscoring the advantages of an explicit 3D representation for robust and interpretable generalization. Further examples are shown in the supplementary.

## 5.4. Preliminary real-world results

Due to the lack of passive multi-view real-world video datasets, we report preliminary results on action-conditioned scenarios. We evaluate our method on the HO-Cap dataset of (Wang et al., 2024b), which contains 64 trajectories captured using 7 synchronized cameras. Since the dataset is too small to train a multi-view encoder from scratch, we replace MVSplat with a pretrained DepthAnything3 (giant-large-1.1) model (Lin et al., 2025a), fine-tuning its depth DPT head and adding a second DPT head to predict latent features. We train at 15 FPS using only 3 camera views that cover the full scene and otherwise resort to the same training parameters as previously discussed.

To handle non-overlapping camera fields of view, we compute the image reconstruction loss solely across the context camera set. Specifically, each camera is reconstructed from the remaining context cameras (e.g., camera b is novel with respect to camera a), but we do not supervise reconstruction on additional target cameras that are novel to all context views. Initializing from a strong pretrained model makes this possible: per-view reconstructions are accurate enough to support novel-view rendering. For action conditioning, we incorporate sparse 3D hand keypoints with learned embeddings, which are otherwise treated identically to the remaining particles in the scene representation.

As shown in (Figs. 13 and S12–S13), despite the limited size

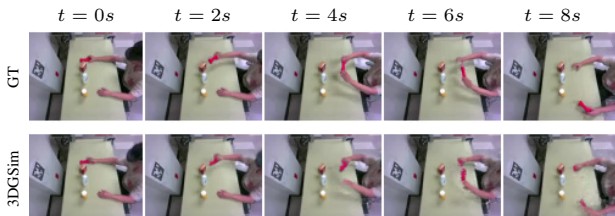

*Figure 13.* **Qualitative results on real-world scenarios.** We present an 8-second (120-step) HO-Cap rollouts illustrate that the model can predict complex dynamics, such as object handovers.

of the dataset, our model is able to simulate long-horizon interaction dynamics over 120 steps using only initial frames and action conditioning. The results include complex scenarios such as object handovers between hands (Fig. S12) as well as single-hand object manipulation (Fig. S13). We additionally report a representative failure case in (Fig. S15). A quantitative analysis of the predictions is provided in (Appendix F.3), and videos corresponding to all examples are available in the supplementary material.

## 6. Discussion

We introduced 3DGSim, a fully differentiable 3D Gaussian simulator that learns directly from multi-view RGB video. 3DGSim integrates inverse rendering, dynamics prediction, and novel-view video synthesis within a single end-to-end learnable system. Given that 3DGSim pioneers an unexplored direction for 3D particle-based simulation, future work will further explore action conditioningto enable large-scale validation on real-world multi-view datasets. for passive phenomena. The dependence on multi-view inputs can be further mitigated by recent advances in monocular inverse rendering (Wang et al., 2025; Murai et al., 2024). Additionally, while occlusions are not explicitly modeled, they are partially addressed by the dynamics module and may be further improved through point completion techniques.

**Spatial Causality.** In 3DGSim, interactions are restricted to those between spatially grounded particles, which ensures that the simulation adheres to realistic physical dynamics. This contrasts with 2D pixel-based video generation models, where apparent dynamics often emerge from the generative flexibility of image-space synthesis. The 3D formulation thus brings advantages such as spatial consistency, object permanence, and robustness to out-of-distribution inputs, as exemplified by our generalization tests. However, it introduces certain compromises: while 2D predictors can effortlessly repurpose pixels to synthesize novel content, e.g., fabricating unseen objects from generative priors, 3D particle models inherit stricter structural constraints, making it difficult to dynamically create or destroy particles in a learnable manner. This highlights a tradeoff between the stability and interpretability provided by 3D spatial causality and the generative freedom unlocked by 2D video models.

**Toward Vision-Language-driven Simulation.** Integrating language embeddings offers a promising avenue for enriching particle-based simulations. Unsupervised vision models such as Dino v3 (Siméoni et al., 2025) provide informative priors about object properties, enabling more structured and semantically aware predictions. We envision 3DGSim as a step toward scalable simulators that can learn physical interactions from both visual and textual modalities, ultimately supporting a more nuanced robotic understanding of complex real-world dynamics.

## Acknowledgements

We thank Radek Daněček and Omid Taheri for valuable guidance and discussions on challenges in 3D computer vision. We are additionally grateful to Wieland Brendel for insightful conversations during the early ideation phase that helped shape this project. We also thank the Max Planck Institute for Intelligent Systems for providing compute resources that supported this research.

## Impact Statement

This paper presents work whose goal is to advance the field of Machine Learning. There are many potential societal consequences of our work, none which we feel must be specifically highlighted here.

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

# 3DGSim

# Supplementary Material

In this Supplementary Material, we provide additional implementation details in (Sec. A), extended experimental results in (Sec. B), ablations in (Sec. C), a discussion of prior work in (Sec. D) and the rationale for image based evaluation metrics in (Sec. E). The remainder of the supplementary material (Sec. F) contains additional rollout visualizations for different scenarios.

## A. Implementation Details

### A.1. Unprojecting Pixel-Aligned Features via FiLM Conditioning

To transform the view-dependent pixel-aligned features $\hat{f}'_i$ into a consistent 3D representation, we use a multilayer perceptron (MLP) with Feature-wise Linear Modulation (FiLM). FiLM conditioning enables the MLP to adapt its processing of $\hat{f}'_i$ based on geometric context, such as camera viewpoint and depth.

Specifically, FiLM computes a scale and bias using a conditioning network $\gamma$ that takes as input a geometric context vector $\mathbf{x}_i$, which includes depth, density, and pixel shift, as well as the Plücker ray encoding $\mathbf{r}_i = [\mathbf{o}_i \times \mathbf{d}_i \mid \mathbf{d}_i,]$, where $\mathbf{o}_i$ and $\mathbf{d}_i$ denote the origin and direction of the viewing ray:

$$\text{scale}_i, \text{ bias}_i = \gamma(\mathbf{x}_i, \mathbf{r}_i). \tag{S1}$$

These parameters modulate the activations of the MLP processing $\hat{f}'_i$ through FiLM layers:

$$f_i = \text{MLP}(\hat{f}'_i; \text{scale}_i, \text{bias}_i), \quad \text{FiLM}(h) = \text{scale}_i \odot h + \text{bias}_i, \tag{S2}$$

where $h$ denotes a hidden activation and $\odot$ is element-wise multiplication.

This setup allows the network to unproject image-aligned features into canonical 3D space while respecting scene geometry and view direction.

### A.2. PTv3: Scalable point cloud transformations

**Point cloud serialization**   At the core of PTv3 lies "point cloud serialization", an algorithm that transforms an unstructured point cloud into ordered points. This process begins by discretizing 3D space into a uniform grid of points. As illustrated in (Fig. 6), these point are then connected using a space-filling curve – a path that traverses each grid point exactly once while preserving spatial proximity as much as possible. Each point $p_i$ is assigned an integer code $s_i$, representing its position within a space-filling curve, via the mapping

$$s_i = \phi^{-1}(\lfloor p_i/G \rfloor) \tag{S3}$$

with $\phi^{-1} : \mathcal{Z}^N \mapsto \mathcal{Z}$ and grid size $G \in \mathbb{R}$. The points in the clouds are then ordered by their respective code $s_i$, yielding a serialized point cloud (SPC)

$$S_i(t_k) = \{(s_i, \tilde{g}_i)\}_{i=1}^{N_k}. \tag{S4}$$

While this approach may not preserve local connectivity as precisely as kNN groupings, (Wu et al., 2024b) emphasizes that the slight loss in spatial precision is outweighed by a significant gain in computational efficiency. To obtain diverse spatial connections between points, PTv3 shuffles between four different space filling curve patterns to obtain SPCs from which patches are computed and varies the patch computation through integer dilation.

**Patch grouping**   PTv3 partitions the SPC $S_i(t_k)$ into equally sized patches and applies self-attention within each patch. To ensure divisibility, patches that do not align with the specified size are padded by borrowing points from neighboring patches.

**Conditional embeddings and patch attention**   Besides the computational efficiency of SPC over kNN, its main advantage lies in the compatibility with standard dot-product attention mechanisms. To understand this, we first examine how the standard PTv3 architecture computes particle-wise predictions from a single point cloud at a single time step. The process

begins by extracting particle-wise embeddings $E_i$ for each serialized point $(s_i, \tilde{g}_i)$ using a sparse convolutional neural network (CNN). Next, the embedded SPCs are progressively down sampled via grid pooling before being grouped and shuffled into patch pairs. Conditional positional embeddings (xCPE) are then added to the embeddings, followed by layer normalization and a patch-wise attention layer predicting the change in the embeddings $\Delta E_i$. The pooling, patch shuffling, and attention blocks are arranged in a U-net (Ronneberger et al., 2015) like architecture that first reduces the size of the SPCs in an encoding step and then mirrors this architecture. In 3DGSim, the final layer of the dynamics model predicts the change in the particle positions $\Delta p_i$ and the change in their features $\Delta f_i$.

### A.3. Architectures

**Feature Encoding Network**  In (Section 4.2) we describe the feature encoding network that transforms pixel-aligned features into view-independent the latent features. To regress the FiLM conditioning we use a 2-layer CNN with GELU activation, kernel of size 3 and channel dimensions $(10, 20)$. The first dimension also matches the size of the conditioning vector.

**Particle Wise MLP**  At the end of the dynamics model, the embedding of each particle is mapped back to the particle latent features. For that we use an MLP with shapes $(128, 128)$ and GELU activation between each layer.

### A.4. Simulation Speed

Simulation speed is critical for robotics applications. Traditional simulators (FEM, MPM, PBD) typically employ small integration timesteps. Learned approaches enable larger timesteps, allowing 3DGSim to simulate elastic cloth at $42$ FPS and rigid dynamics at $12$ FPS, with inference speeds of $\sim 16$ FPS (4 past steps) and $\sim 20.1$ FPS (2 past steps), using under $20$ GB VRAM on an H100 GPU and achieving *near real-time speeds*.

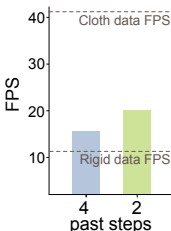

*Figure S1.* Prediction vs simulation FPS.

## B. Extended Experimental Results

(Table. S1) summarizes the quantitative performance of 3DGSim, Cosmos, and Spring-Gaus. Further discussion of Spring-Gaus is presented in (Sec. B.1), and (Sec. B.2) analyzes inference with variable-length temporal inputs.

Across all experiments, we report PSNR, SSIM, and LPIPS as frame-level visual metrics, together with Fréchet Video Distance (FVD) to evaluate temporal video quality.

### B.1. Comparison with Spring-Gaus

We evaluated Spring-Gaus (Zhong et al., 2024) with default parameters on 3 test-set trajectories per object, providing ground-truth ground height and object masks as required inputs. For velocity-estimation we use the first 4 frames. Several caveats apply to this comparison. First, Spring-Gaus only supports contact via a boundary condition against a known ground plane and does not handle general particle collisions, ruling out cloth dynamics and multi-object interaction. Second, its estimated parameters (per-anchor stiffness, rest lengths, connectivity, ..) are tied to a specific point cloud topology derived from random sampling of the static reconstruction. Since there is no correspondence between anchor points across different trajectories, parameters cannot transfer, forcing optimization and testing on the same trajectory (length 84 for elastic, 64 for rigid). Third, the method does not simulate visual effects such as shadows or lighting, which 3DGSim handles through its learned implicit simulator. Results reflect these constraints. Soft dynamics are better estimated, though plasticity remains uncaptured. Rigid bodies recover appropriately stiff springs, but bounciness leads to poor overall accuracy. In the full-scene setting, the lack of visual simulation accounts for an additional 2-3 dB PSNR reduction.

| Dataset | Model | PSNR (future) ↑ | PSNR (past) ↑ | SSIM(future) ↑ | LPIPS(future) ↓ | FVD ↓ |
|---|---|---|---|---|---|---|
| Elastic | 3DGSim 4-12 latent | **33.15 ± 3.51** | 34.42 ± 2.34 | **0.97 ± 0.02** | **0.02 ± 0.01** | 91.3 |
| | 3DGSim 2-12 latent | 32.05 ± 3.48 | 35.98 ± 1.96 | 0.96 ± 0.02 | **0.03 ± 0.01** | **64.0** |
| | 3DGSim 4-12 explicit | 29.69 ± 1.75 | 39.92 ± 3.21 | 0.92 ± 0.01 | 0.09 ± 0.01 | 215.0 |
| | 3DGSim 2-12 explicit | 29.98 ± 1.60 | 40.62 ± 3.09 | 0.92 ± 0.01 | 0.11 ± 0.01 | 246.8 |
| | CosmosFT | 26.97 ± 4.22 | – | 0.82 ± 0.03 | 0.07 ± 0.03 | 156.4 |
| | Spring-Gaus † | 24.13 ± 3.65 | 37.20 ± 2.08 | 0.94 ± 0.02 | 0.08 ± 0.02 | |
| | Spring-Gaus (obj only) | 27.13 ± 3.33 | 37.20 ± 2.08 | 0.97 ± 0.01 | 0.04 ± 0.01 | |
| Cloth | 3DGSim 4-8 latent | **26.46 ± 2.62** | 34.89 ± 2.23 | **0.88 ± 0.03** | 0.09 ± 0.02 | 117.5 |
| | 3DGSim 2-8 latent | 26.23 ± 2.24 | 35.27 ± 1.92 | **0.88 ± 0.02** | **0.08 ± 0.02** | **78.4** |
| | 3DGSim 4-8 explicit | 24.40 ± 1.68 | 39.62 ± 2.01 | 0.86 ± 0.02 | 0.12 ± 0.02 | 112.7 |
| | 3DGSim 2-8 explicit | 18.48 ± 1.50 | 32.91 ± 0.80 | 0.69 ± 0.04 | 0.33 ± 0.02 | 1143.3 |
| | CosmosFT | 23.12 ± 0.69 | – | 0.72 ± 0.02 | 0.15 ± 0.03 | 134.5 |
| | Spring-Gaus | – | – | – | – | – |
| | Spring-Gaus (obj only) | – | – | – | – | – |
| Rigid | 3DGSim 4-12 latent | **28.35 ± 2.70** | 32.86 ± 1.51 | **0.91 ± 0.03** | **0.08 ± 0.02** | 297.4 |
| | 3DGSim 2-12 latent | 27.85 ± 2.28 | 32.91 ± 1.59 | 0.90 ± 0.02 | 0.09 ± 0.02 | **232.9** |
| | 3DGSim 4-12 explicit frozen | 25.50 ± 1.59 | 35.55 ± 1.70 | 0.85 ± 0.02 | 0.17 ± 0.02 | 372.6 |
| | 3DGSim 2-12 explicit | 24.93 ± 1.67 | 34.80 ± 1.68 | 0.85 ± 0.03 | 0.16 ± 0.02 | 391.0 |
| | CosmosFT | 27.31 ± 2.57 | – | 0.70 ± 0.06 | 0.11 ± 0.04 | 351.5 |
| | Spring-Gaus † | 23.85 ± 1.01 | 33.89 ± 1.39 | 0.89 ± 0.02 | 0.13 ± 0.03 | |
| | Spring-Gaus (obj only) | 27.39 ± 1.47 | 35.81 ± 1.68 | **0.94 ± 0.01** | 0.08 ± 0.02 | |

*Table S1.* Detailed quantitative results across all models and datasets. 3DGSim 4-12 latent uses a 4-step conditioning window and is trained on 12 future steps. Spring-Gaus † includes a static ground reconstruction for rendering; lower scores are mainly attributed to unresolved appearance effects such as shadows and lighting. Furthermore, Spring-Gaus cannot model cloth dynamics due to limited collision handling capabilities.

## B.2. Variable-length input at inference time

We evaluate variable-length input windows at inference time on the elastic dataset. Performance peaks at 4 past frames as this matches the training configuration. With fewer steps (1–2), the model lacks sufficient temporal context. With more steps (8), performance degrades because TEM only learns positional embeddings for the first two hierarchy levels (matching 4 input steps); deeper levels are engaged with untrained embeddings, corrupting feature representations. That said, even in these cases the dynamics remain qualitatively plausible, with degradation appearing mainly as reduced post-collision accuracy. Training with variable-length input windows is an interesting direction for future work.

| past frames | PSNR (future) ↑ | SSIM ↑ | LPIPS ↓ | FVD ↓ |
|---|---|---|---|---|
| **4** | **33.15** | **0.97** | **0.02** | **91.3** |
| 8 | 28.60 | 0.95 | 0.04 | 242.2 |
| 2 | 27.27 | 0.92 | 0.08 | 387.8 |
| 1 | 22.92 | 0.87 | 0.14 | 959.2 |

*Table S2.* Evaluation of *3DGSim 4-12 latent* under variable-length input windows.

## C. Ablations

For the ablation studies of 3DGSim, only the elastic dataset is used. The default configuration uses the latent representation, 4-step past conditioning, 12-step future rollouts, 4 input views, 5 target views, and a total of 12 cameras for training, unless otherwise specified. Any deviations from these parameters are explicitly stated or made clear within the context of the respective ablation.

This analysis provides detailed insights into design and strategic choices that inform future improvements of our approach.

## C.1. Rollout Length

| Rollout Length | PSNR Future | PSNR Past |
|---|---|---|
| 2 steps | 26.43 ± 3.48 | 35.70 ± 2.51 |
| 4 steps | 28.83 ± 4.96 | 35.25 ± 2.53 |
| 8 steps | 30.64 ± 3.23 | 33.86 ± 2.17 |
| **12 steps** | **33.15 ± 3.51** | **34.55 ± 2.26** |

*Table S3.* Rollout Length

We evaluate the influence of prediction rollout length during training (2, 4, 8, and 12 steps). Consistent with expectations, results improve significantly as the rollout length increases, reaching a peak performance at 12 steps with a PSNR Future of $33.15 \pm 3.51$. Extending the number of rollout steps enhances the model's predictive capability but leads to significant memory requirements. Employing regularization methods such as random-walk noise injection or diffusion techniques or even other modality (see later) can help reduce the required rollout steps.

## C.2. Camera Setup

| Setup | PSNR Future | PSNR Past |
|---|---|---|
| Explicit 3 views out of 6 | 21.02 ± 1.78 | 16.86 ± 0.83 |
| **Latent 3 views out of 6** | **31.60 ± 3.09** | **32.55 ± 2.12** |
| **Latent 4 views out of 12** | **33.15 ± 3.51** | **34.55 ± 2.26** |

*Table S4.* Camera Setup (85k steps)

To approximate a realistic scenario suitable for real-world deployment, we investigate performance with reduced camera setups. Interestingly, the latent representation models achieve robust performance even when trained with 3 views out of 6 total cameras (PSNR 33.15), whereas explicit representation models degrade significantly (PSNR drops to 21.02) due to convergence of the encoder to poor local minima. This local minimum manifests as camera-specific overfitting, where the model erroneously predicts particle arrangements forming planar, screen-aligned shapes. While this artificially reduces the training loss of target viewpoints, it undermines the true representation quality and disrupts convergence to a consistent 3D reconstruction. Further restricting the setup to only 2 views out of 4 cameras results in unsuccessful training for both latent and explicit models, indicating that very limited camera setups demand careful placement or preliminary encoder pre-training, which should be investigated in a future work.

## C.3. Segmentation Masks

| Segmentation | PSNR Future | PSNR Past |
|---|---|---|
| Without masks | 32.66 ± 3.43 | 39.08 ± 3.18 |
| With masks | 33.15 ± 3.51 | 34.55 ± 2.26 |

*Table S5.* Segmentation Masks

While our final models use segmentation masks for static objects (e.g., ground surfaces), we explore training without these masks to test model reliance on explicit segmentation. We find only a slight reduction in performance (32.66 vs. 33.15), demonstrating the models' capability to implicitly infer static scene regions directly from raw RGB inputs. Thus, explicit segmentation masks are helpful but not strictly essential.

## C.4. Modality Configurations

| Modality | PSNR Future | PSNR Past |
|----------|-------------|-----------|
| 4-1-2-6 | 32.59 ± 3.22 | 33.39 ± 2.21 |
| 4-4-1-3 | 31.98 ± 3.78 | 34.03 ± 2.39 |
| 4-1-1-12 | 33.15 ± 3.51 | 34.55 ± 2.26 |

*Table S6.* Input Modality

Different input modality configurations were tested. Here, we adapt the notation "a-b-c-d", each varying the temporal span of input conditioning $a$ is the number of particle frames representing the state, $b$ indicates the number of backward frames used to predict $c$ future steps, and total rollout steps during training $d$, leading to $b \cdot c \cdot d$ rollout steps per training step. Both variants ("4-1-2-6", "4-4-1-3" attain competitive performance (PSNR of 32.59 and 32.27 respectively), significantly reducing the computational load compared to longer standard rollouts. This highlights promising avenues for future investigation, emphasizing balance between computational efficiency and performance quality.

## C.5. Grid Resolution

| Grid Size | PSNR Future | PSNR Past |
|-----------|-------------|-----------|
| 0.002 | 25.39 ± 2.95 | 32.28 ± 2.68 |
| **0.004** | **33.15 ± 3.51** | **34.55 ± 2.26** |
| 0.008 | 25.40 ± 3.84 | 30.78 ± 2.63 |
| 0.0012 | 24.06 ± 3.53 | 31.74 ± 2.56 |

*Table S7.* Grid Resolution.

We test a series of grid resolutions (0.002, 0.004, and 0.008), observing optimal results at 0.004 with a PSNR of 33.15. Both higher (0.008) and finer resolutions (0.002) degrade performance, suggesting an optimal balance achieved at 0.004 between detail preservation and computational complexity for out scene size.

## C.6. Temporal Merger

| Temporal Merger Setup | PSNR Future | PSNR Past |
|-----------------------|-------------|-----------|
| [1,1,2,2..] with embedding | 27.55 ± 3.22 | 31.68 ± 2.60 |
| [1,1,4,..] with embedding | 26.79 ± 2.94 | 32.21 ± 2.73 |
| [1,2,2,..] without embedding | 25.07 ± 3.22 | 31.16 ± 2.59 |
| **[1,2,2,..] with embedding (120k)** | **33.15 ± 3.51** | **34.55 ± 2.26** |
| [1,1,1,2,2] with embedding | 18.87 ± 1.50 | 18.09 ± 1.45 |
| [1,4,..] with embedding | 18.19 ± 1.29 | 18.33 ± 1.48 |

*Table S8.* Temporal Merger. The models are trained for 80k steps if not specified otherwise.

We experiment with various temporal merging strides for each encoder stage combined with embedding options of timestep position encoding. Since we only train with 4 past steps, the strides ".." don't influence the results. After 80k iterations, results clearly indicate two critical factors for success: the use of learned positional embeddings and timing of merging operations. Optimal results occur when merging temporal information only after early spatial processing stages ([1,2,2..]), whereas early or too-late merging drastically diminishes performance. Poor outcomes with late merging likely arise due to spatial pooling operations that dilute vital temporal distinctions before merging.

# D. Discussion of Prior Work

To clarify the scope of our contributions, we distinguish 3DGSim from methods focused on dynamic scene reconstruction, and then contrast it with approaches that augment reconstructed 3D scenes with simulation capabilities.

## D.1. Distinction from dynamic scene reconstruction

Here, we elucidate the fundamental differences between our approach, 3DGSim, and dynamic scene reconstruction methods such as 4DGS (Wu et al., 2024a) and DeformableGS (Bae et al., 2024).

**Temporal Characteristics:** Dynamic scene reconstruction techniques, including 4DGS and DeformableGS, aim to reconstruct a temporally-varying 3D representation from a complete set of video frames, encompassing past, present, and future data. These methods are inherently temporally *non-causal*, as they leverage the entire video sequence to infer and optimize a 4D (3D + time) representation of the observed events. Their primary function is to interpolate or reconstruct what has already occurred within the video.

**Predictive vs. Reconstructive Nature:** In stark contrast, 3DGSim is designed for temporally *causal prediction*. Given a set of scene representations from a few past timesteps, our model's objective is to predict the future evolution of the scene. This prediction is made without any access to future video frames. This predictive capability necessitates that the model develops an understanding of the underlying physics governing the scene's dynamics.

## D.2. Comparison to PhysGaussian, PhysDreamer, and Spring-Gaus

PhysGaussian (Xie et al., 2023), PhysDreamer (Zhang et al., 2024b), Spring-Gaus (Zhong et al., 2024) reconstruct 3D representations from multi-view images of static scenes and then simulate mesh-node dynamics via the Material Point Method (MPM) or an analyitcal spring-simulator. While these methods are effective for VR and per-scene content creation, a direct comparison to 3DGSim is not feasible for several reasons:

**PhysGaussian does not learn dynamics.** PhysGaussian reconstructs a 3D representation from a *static* scene and simulates particles with MPM. In our experiments, training images are generated using a combination of MPM simulators. One could attempt to tune the PhysGaussian MPM simulator to match the data-generating simulation, but hand-tuning MPM parameters to match the motion of another simulation is notoriously difficult. In contrast, 3DGSim learns to simulate object dynamics directly from video without requiring physical priors. *While our method can be extended to learn different material modalities and collisions, it is unclear how to tackle this problem with MPM.*

**PhysDreamer performs per-scene optimization.** Our method is a feed-forward world model trained once and reused across scenes. By contrast, PhysDreamer requires training a separate model for each trajectory.

**PhysDreamer does not simulate collisions.** As the authors note: "In this work, we restrict our scope to elastic objects without collisions." In our experiments, 3DGSim accurately simulates both elastic and rigid collisions.

**In PhysDreamer (and MPM in general), boundary conditions must be defined by hand.** Static parts of objects are manually set to be static. 3DGSim learns constrained dynamics directly from videos.

**MPM requires very small simulation steps, making full-trajectory backpropagation impractical.** For instance, in PhysDreamer the timestep is $\Delta t = 1 \times 10^{-4}$ with 768 sub-steps per frame. Training is split into two stages: 1) optimizing initial velocities on a few frames, 2) estimating material parameters with fixed velocities. In the second stage, gradients are truncated to flow only one frame backward to avoid explosion/vanishing gradients. As the authors state (Zhang et al., 2024b):

> "Rather than optimizing the material parameters and initial velocity jointly, we split the optimization into two stages for better stability and faster convergence. In particular, in the first stage, we randomly initialize the Young's modulus for each Gaussian particle and freeze it. We optimize the initial velocity of each particle using only the first three frames of the reference video. In the second stage, we freeze the initial velocity and optimize the spatially varying Young's modulus. During the second stage, the gradient signal only flows to the previous frame to prevent gradient explosion/vanishing."

In contrast, 3DGSim jointly learns 3D reconstruction and dynamics simulation in a *fully end-to-end* manner, with *backpropagation through entire trajectories*, without requiring staged training or gradient truncation. Its transformer-based architecture enables significantly larger timesteps. Empirically, 3DGSim achieves accurate long-range predictions while being *1–2 orders of magnitude faster to train and simulate*.

**Spring-Gaus.** Since we were requested a comparison with Spring-Gaus (Zhong et al., 2024), we have taken the time to run Spring-Gaus on our benchmark. The full results and discussion can be found in Sec. B.1. We remain of the opinion that this comparison is not fully fair due to the substantial differences in problem setup outlined above: Spring-Gaus requires ground-truth scene constraints, optimizes and tests on the same trajectory, cannot handle cloth or particle-collisions in general, and does not simulate visual effects.

Nonetheless, the results show that 3DGSim outperforms Spring-Gaus on future prediction across all metrics despite requiring no privileged inputs or per-trajectory optimization. We hope this additional experiment addresses remaining concern regarding baseline comparisons .

**Summary:** The objective of methods like 4DGS and DeformableGS is scene reconstruction. They are not designed to forecast how a physical scene will evolve. Conversely, 3DGSim is a particle-based simulator that learns physics solely from video, enabling forecasting. Approaches like PhysGaussian and PhysDreamer reconstruct 3D representations from static scenes and then use material-point method (MPM) for simulation, they do not incorporate learning of the dynamics in the same manner. In fact, the authors of PhysGaussian emphasize that their framework is unable to handle collisions, a key aspect of physical simulation.

Our work instead tackles vision-based physics learning, which poses a distinct challenge beyond dynamic 3D scene reconstruction.

## E. Rationale for image-based evaluation metrics

The particles of 3DGSim and the simulator we used to create the datasets are fundamentally different: they differ in number, spatial distribution, coverage (ours include the full scene vs. the simulator tracks only dynamic bodies), and semantics (appearance/latent features vs. physical state). The reconstruction is non-unique, so positional differences do not necessarily indicate dynamics errors: post-hoc registration (e.g. nearest-neighbor matching) would conflate reconstruction with dynamics accuracy. This separation is by design: 3DGSim learns to simulate without GT physical state, keeping it applicable to real-world scenarios. Evaluating against that state penalizes the method for the very property that makes it general.

That said, since MVSplat and DA3 are pixel-aligned depth models (constrained to surfaces), whether multi-layer depth models with dynamics-based regularization could recover volumetric structure is an interesting open question.

## F. Visualizations

In the following sections, we present rollout visualizations across a variety of dynamic scenarios. (Sec. F.1) demonstrates scene editability and compositionality, while (Sec. F.2) showcases generalization to multi-body interactions. (Secs. F.3–F.3) present visualizations on the HO-Cap real-world dataset.

Finally, (Sec. F.4) provides qualitative results on the test set across all dynamic categories for both 3DGSim and CosmosFT. Unless stated otherwise, the first row in each visualization corresponds to the ground-truth rollout. CosmosFT is conditioned on the past 5 frames and the following prompts:

| Prompt |
| --- |
| *A rigid body falling on a circular gray ground* |
| *A soft body falling on a circular-gray-ground* |
| *A rigid body falling on a red rectangular cloth which is fixed on its corners* |

*Table S9.* Prompts used to condition Cosmos and CosmosFT.

**Videos of the predictions below are available in the project page.**

### F.1. Scene Editability and Zero-Shot Generalization

In this section, we present extended qualitative results for scene editing tasks that lie outside the training distribution. Leveraging 3DGSim's explicit 3D particle state, we can directly modify initial conditions, such as removing the ground plane or altering release velocities without retraining.

As illustrated in (Fig. S2), our model demonstrates a robust understanding of gravity and momentum. When the ground is removed, the object correctly continues its downward trajectory. In contrast, the image-based baseline, CosmosFT (Fig. S5), exhibits significant artifacts, often leaving objects suspended in mid-air. This comparison suggests that 3DGSim learns generalized physical laws, whereas 2D baselines tend to memorize image-space correlations. Furthermore, we demonstrate that this generalization extends to other parameter changes, such as increased drop heights (Fig. S3) and parallel multi-object simulations (Fig. S4).

#### F.1.1. 3DGSIM

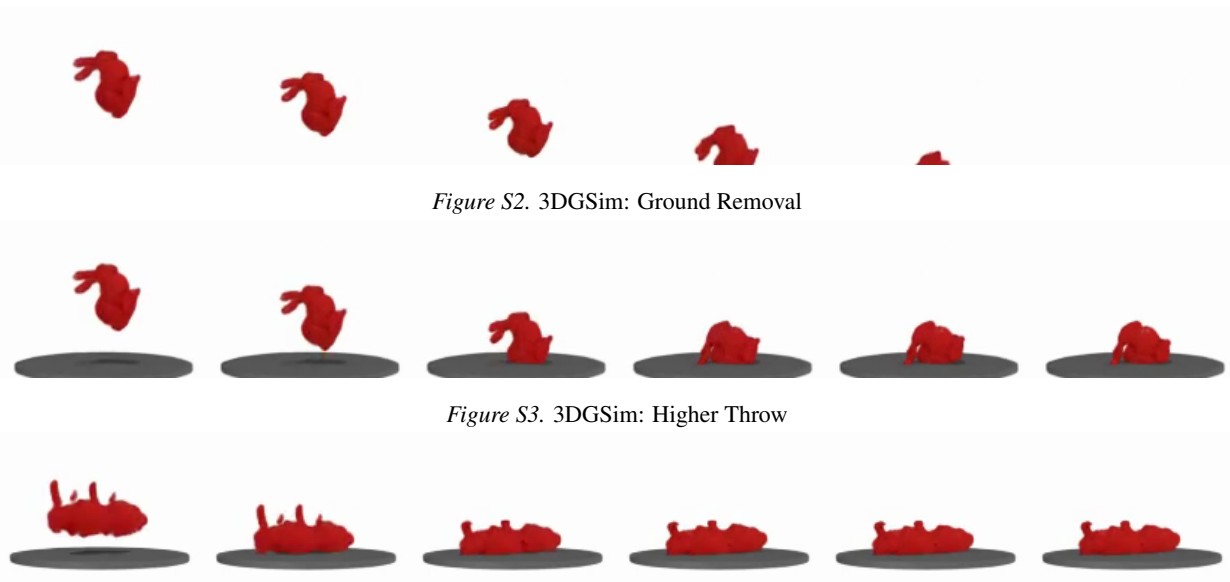

*Figure S2.* 3DGSim: Ground Removal

*Figure S3.* 3DGSim: Higher Throw

*Figure S4.* 3DGSim: Parallel Simulation

#### F.1.2. COSMOSFT

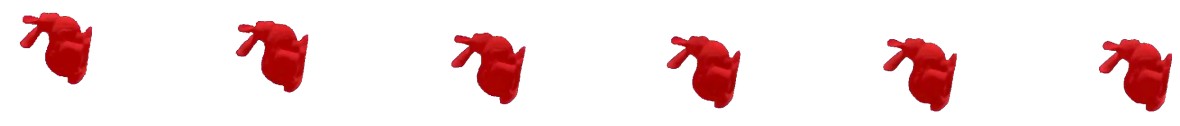

*Figure S5.* CosmosFT: When ground is removed, objects often remain suspended.

## F.2. Generalization to multiple bodies

A key advantage of our particle-based representation is its scalability to multi-body simulations, even when the model was trained exclusively on single-object scenarios. (Fig. S6–S9) showcase the model's ability to handle scenes with two, three, and five concurrent objects.

**3DGSim Observations:** The model successfully maintains the structural integrity of individual objects during collisions. Interestingly, in the high-density scenario (Fig. S8), we observe a realistic "weight" effect where the cumulative pressure of five objects causes slight particle penetration and color shifting at the contact point with the table. This emergent behavior further validates the model's learned spatial-causality.

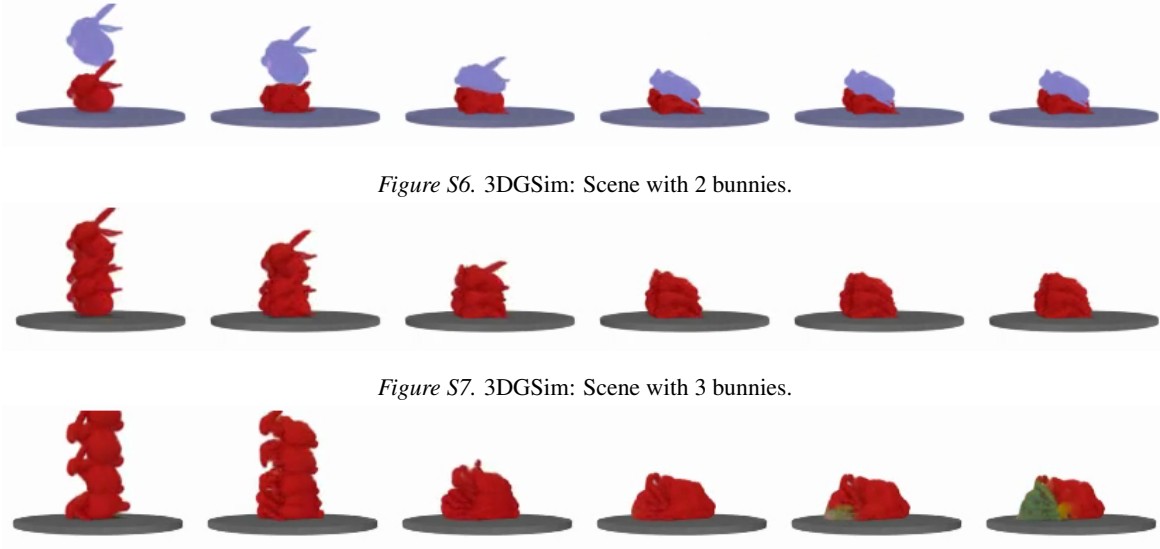

*Figure S6.* 3DGSim: Scene with 2 bunnies.

*Figure S7.* 3DGSim: Scene with 3 bunnies.

*Figure S8.* 3DGSim: Scene with 5 bunnies. Cumulative weight leads to particle-table penetration and color changes.

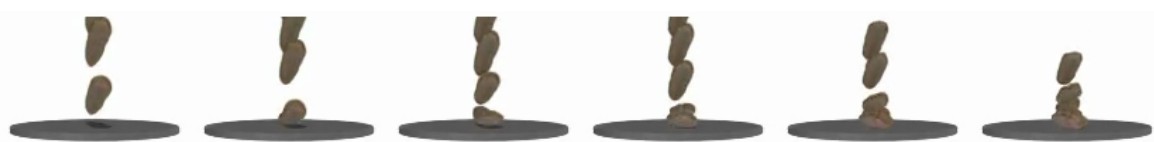

*Figure S9.* 3DGSim: Scene with 5 worms.

**CosmosFT Comparisons:** In contrast, 2D diffusion-based models struggle with multi-body identity. As seen in (Fig. S10), multiple distinct objects tend to "morph" into a single, undifferentiated mass just before contact. Furthermore, (Fig. S11) reveals levitation artifacts, where the model's 2D prior fails to resolve the contact dynamics of overlapping geometries, resulting in physically impossible "floating" states.

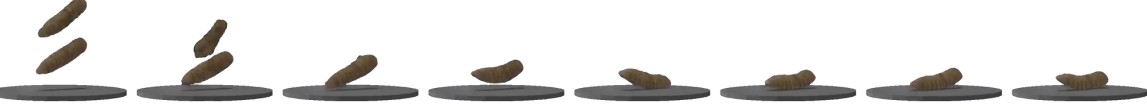

*Figure S10.* Cosmos FT: multiple worms are morphed into one single object before colliding with the ground.

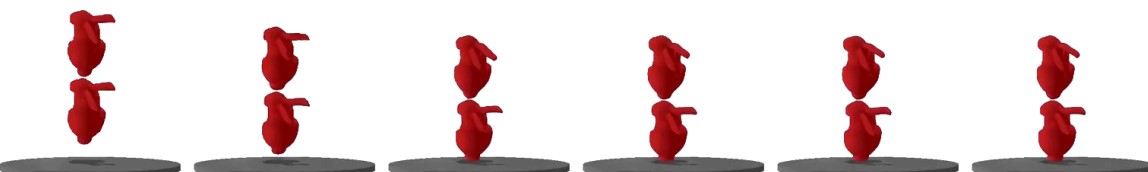

*Figure S11.* CosmosFT: two bunnies levitate above each just before colliding with the ground.

## F.3. Qualitative Analysis on Real-World Dynamics

We present representative rollouts from the HO-Cap dataset, illustrating a diverse range of dynamics. These sequences include both single-hand manipulations in (Fig. S13) and complex bimanual actions, such as hand-overs in (Fig. S12).

Each rollout spans a horizon of 120 steps, corresponding to 8 seconds of simulated time. Despite the complexity of the real-world sensor data, 3DGSim consistently maintains coherent geometry and plausible motion for the majority of the sequence, successfully capturing the subtleties of object transfer and manipulation.

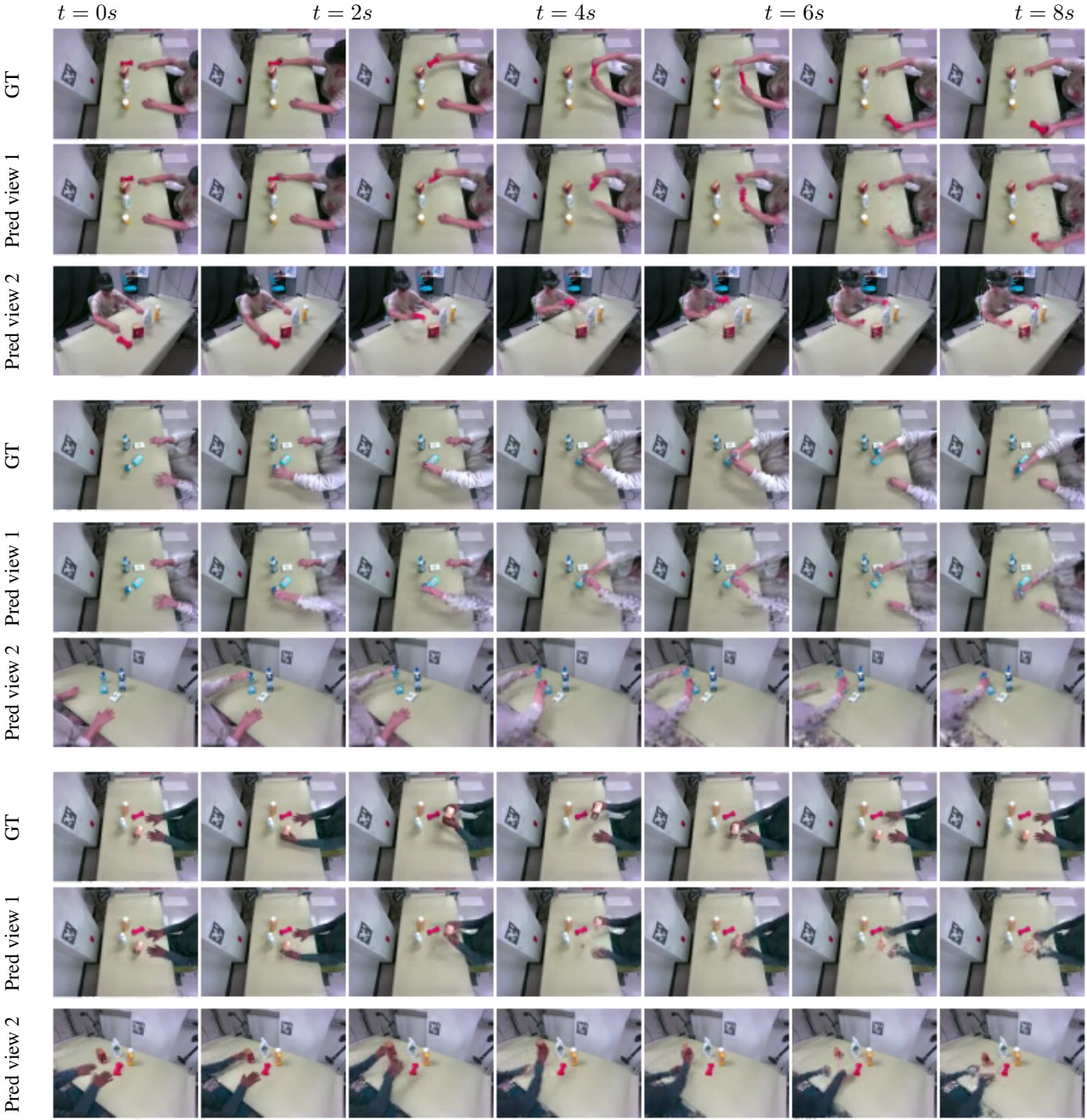

*Figure S12.* **Qualitative results for bimanual manipulation.** These sequences display 8-second (120-step) rollouts from the Ho-Cap dataset, focusing on a complex bimanual handover. The model demonstrates the ability to capture the intricate interaction dynamics as an object is passed from one hand to the other. The Ground Truth (GT) is shown alongside two predicted views to illustrate the model's accuracy in simulating these high-contact transitions.

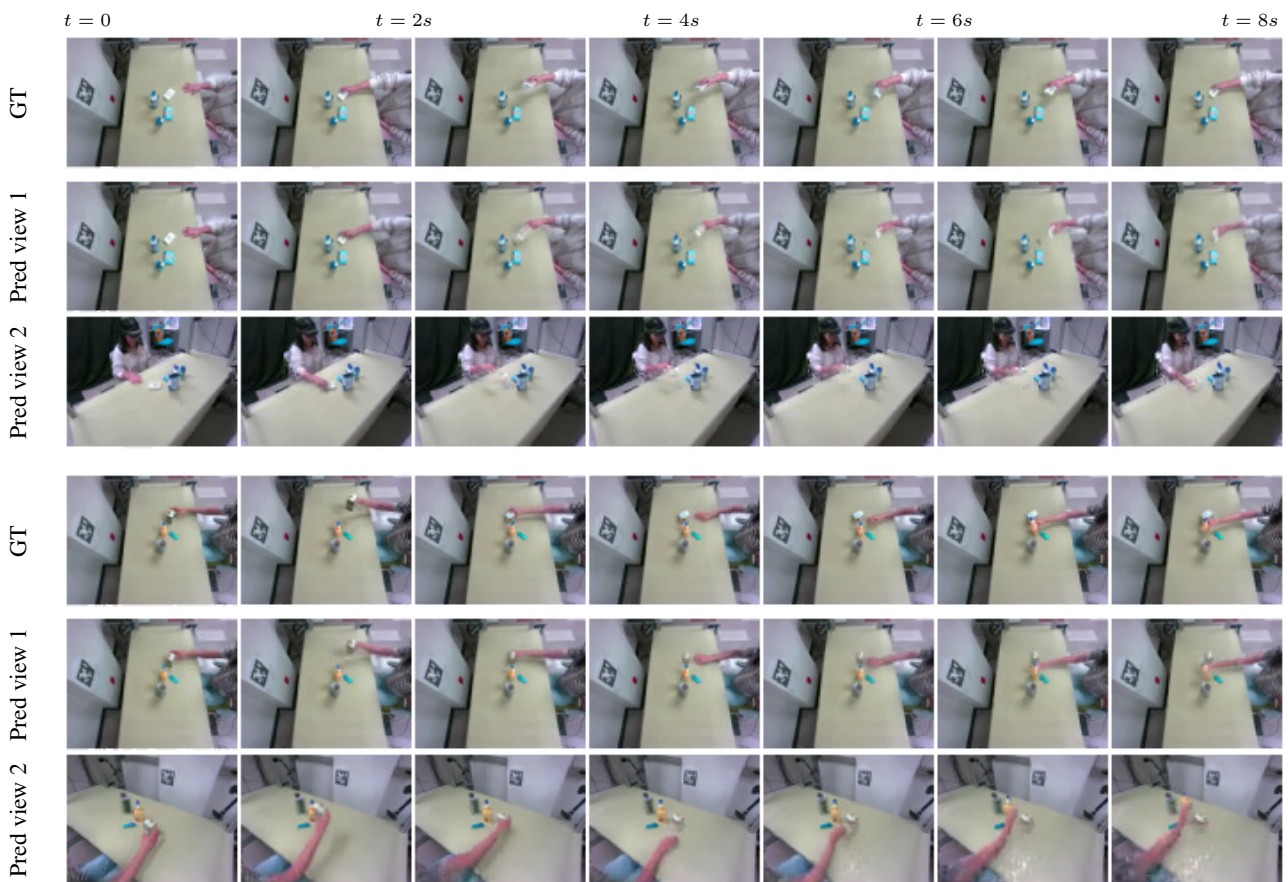

*Figure S13.* **Qualitative results for single-hand manipulation.** We present two representative 8-second (120-step) rollouts illustrating single-hand interaction dynamics within the Ho-Cap dataset. This sequence highlights the model's capacity to simulate stable manipulation of a single object without the complexity of a handover. Comparison between the ground truth and predictions across 120 steps shows the model maintains spatial and temporal consistency throughout the rollout.

**Rollout Stability Analysis on the HO-Cap Dataset**   We evaluate the long-term temporal stability of our method on the HO-Cap dataset by performing a 120-step rollout. (Fig. S14) presents the frame-wise evolution of PSNR, LPIPS, and SSIM. The solid lines represent the mean metric values averaged over the entire dataset, while the shaded regions indicate the standard deviation, highlighting the variability across different test sequences.

We compare our approach against a "constant model" baseline, which simply repeats the last frame of the input window for all future steps. As expected, the baseline (dashed red line) degrades rapidly as the ground truth motion deviates from the initial frame. In contrast, our method (solid blue line) maintains significantly higher stability and perceptual quality throughout the rollout, demonstrating its ability to synthesize plausible dynamics over extended time horizons.

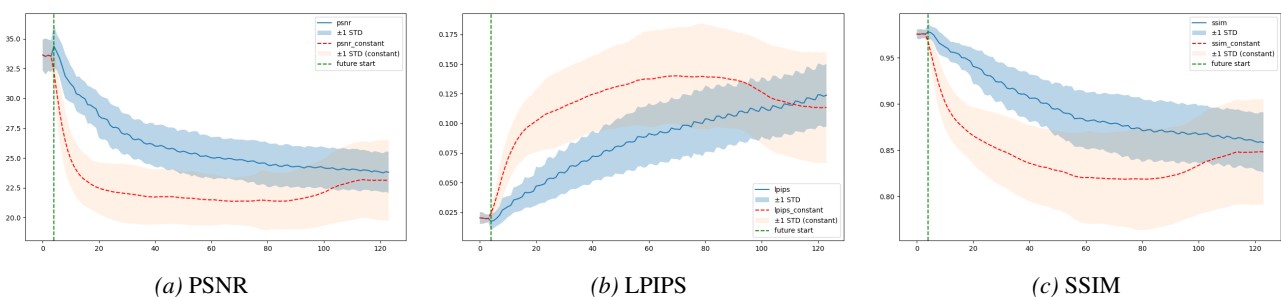

|          *(a)* PSNR          |          *(b)* LPIPS          |          *(c)* SSIM          |

*Figure S14.* Evaluation of metrics over a 120-step rollout. We compare our method against a constant model baseline, which uses the last frame of the input window as the prediction.

**Analysis of Failure Modes** While 3DGSim generates plausible short-term dynamics, we observe characteristic degradation over extended temporal horizons (e.g., $t > 4s$). Due to autoregressive error accumulation, high-frequency details in the hand representation can be lost, causing the underlying particles to blur, drift, or fade into transparency.

However, this failure mode offers an important insight into the model's internal logic. As shown in (Fig. S15), when the hand particles dissipate, the physical interaction (grasping the bottle) immediately ceases. This confirms that the model avoids "hallucinating" object motion and instead relies on learned spatial causality: without the "actuator" (the hand) present, no force is applied to the object.

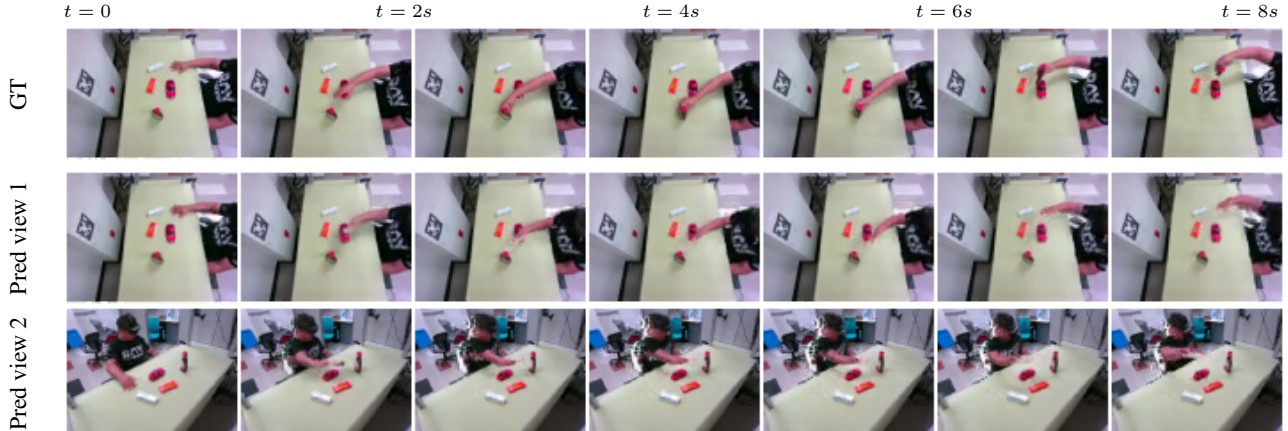

*Figure S15.* **Qualitative results on real-world scenarios.** Over extended horizons, hand blurring can cause particles to drift or fade. However, interactions notably cease when hand particles vanish and the model is unable to pick up the bottle.

## F.4. Visualization from the test set

In this section, we present extensive qualitative results from the test set, covering three distinct physical regimes: cloth interactions, elastic deformation, and rigid body dynamics. For each scenario, we compare the predictions of 3DGSim (bottom row) with the Ground Truth (top row) and the image-based baseline, CosmosFT.

### F.4.1. CLOTH DYNAMICS

( Fig. S16 - S17) illustrate objects with varying geometries (Dragon, Duck, Cow, Devil) dropped onto a suspended cloth fixed at four corners. 3DGSim accurately captures the complex deformation of the fabric as it interacts with the object's geometry. In contrast, CosmosFT often struggles to maintain the structural coherence of the cloth, leading to blurring or incorrect folding patterns.

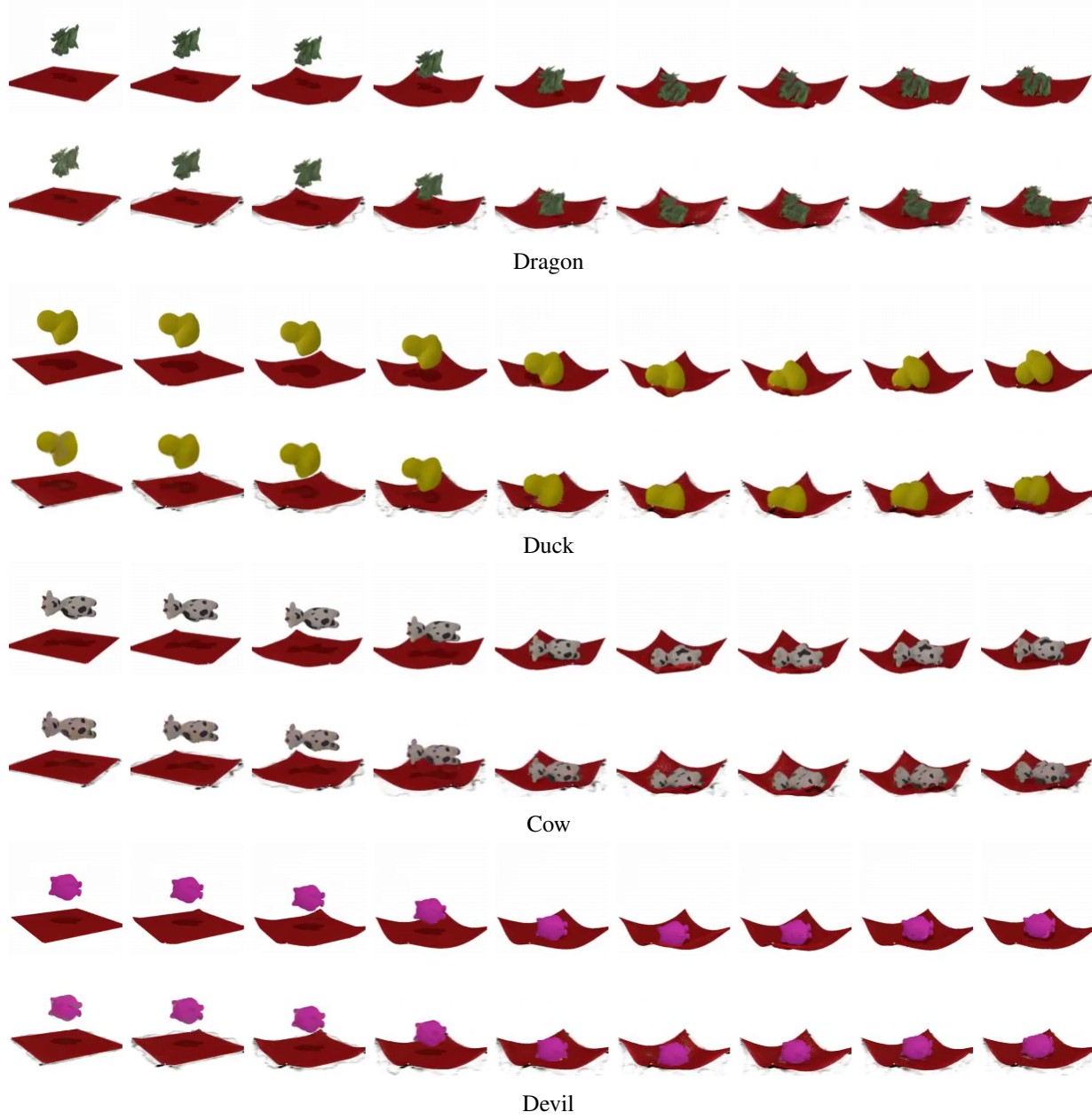

Dragon

Duck

Cow

Devil

*Figure S16.* 3DGSim: Various objects falling on a red cloth with its corners fixed.

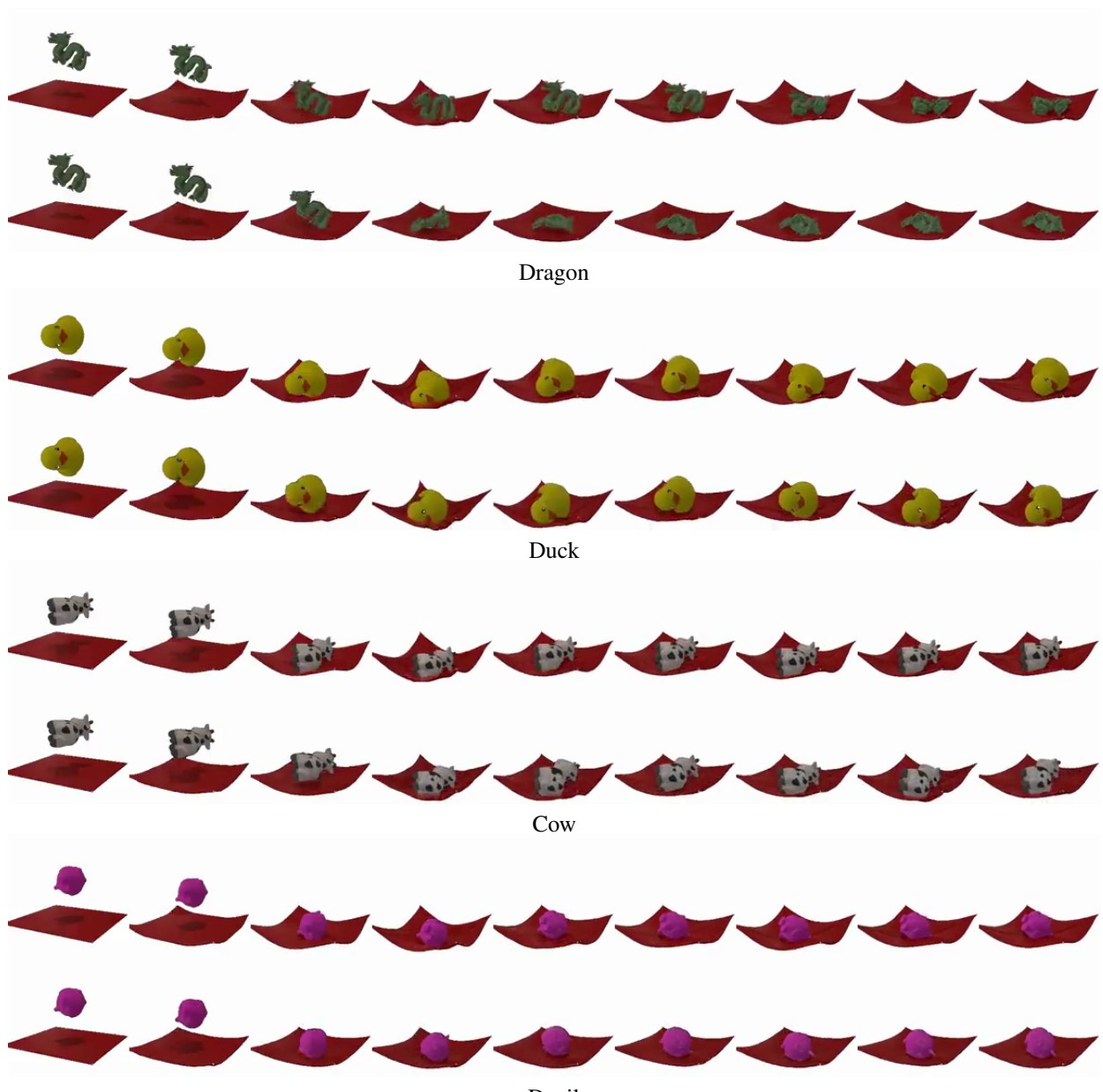

*Figure S17.* CosmosFT: Various objects falling on a red cloth with its corners fixed.

### F.4.2. ELASTIC DYNAMICS

We evaluate soft-body deformation in (Fig. S18 – S19). The sequences depict high-velocity collisions where objects like the "Worm" and "Pig" impact the ground. 3DGSim preserves the volumetric integrity of the objects while simulating plausible compression and rebound.

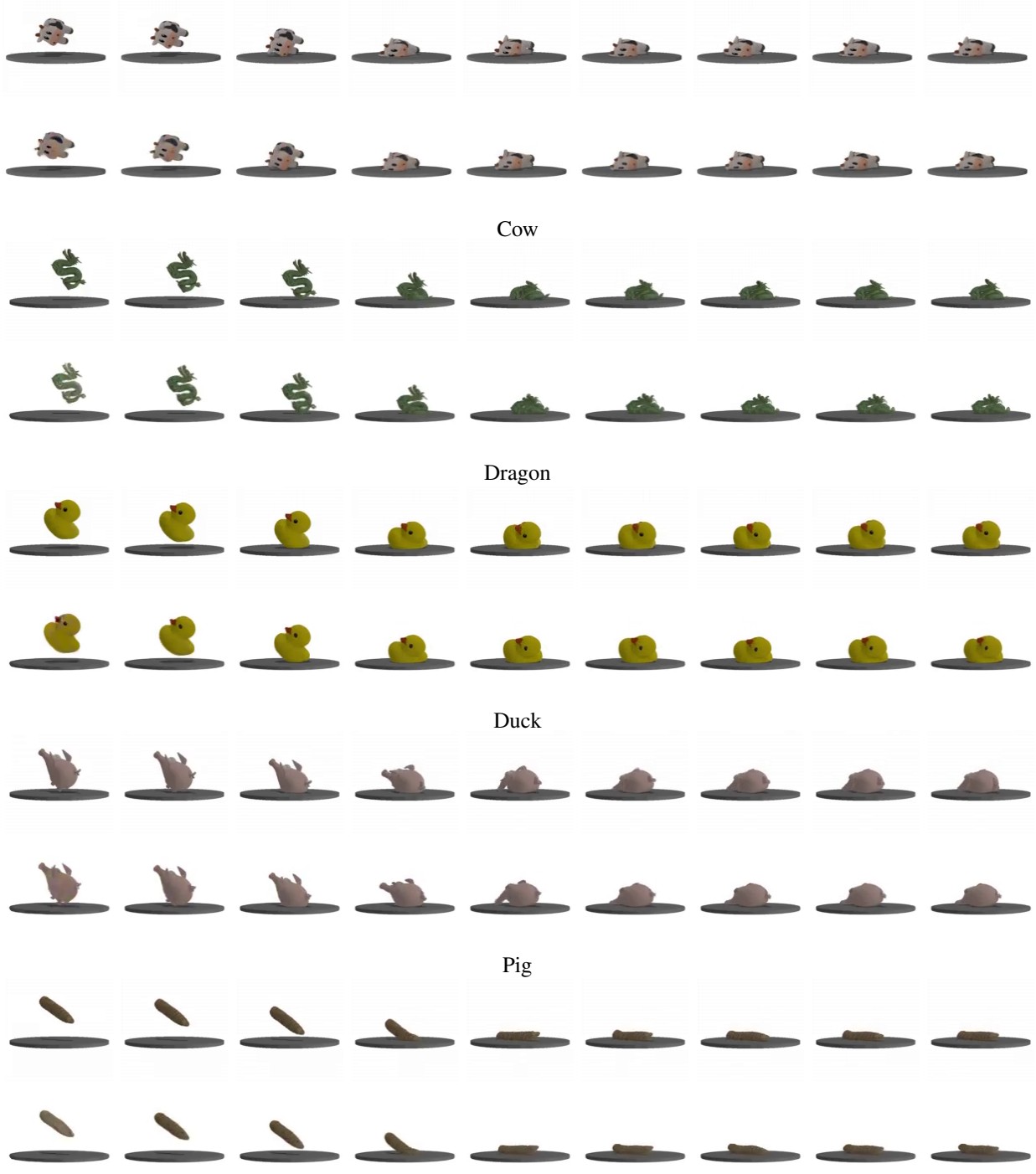

Cow

Dragon

Duck

Pig

Worm

*Figure S18.* 3DGSim: Various elastic simulations for different objects.

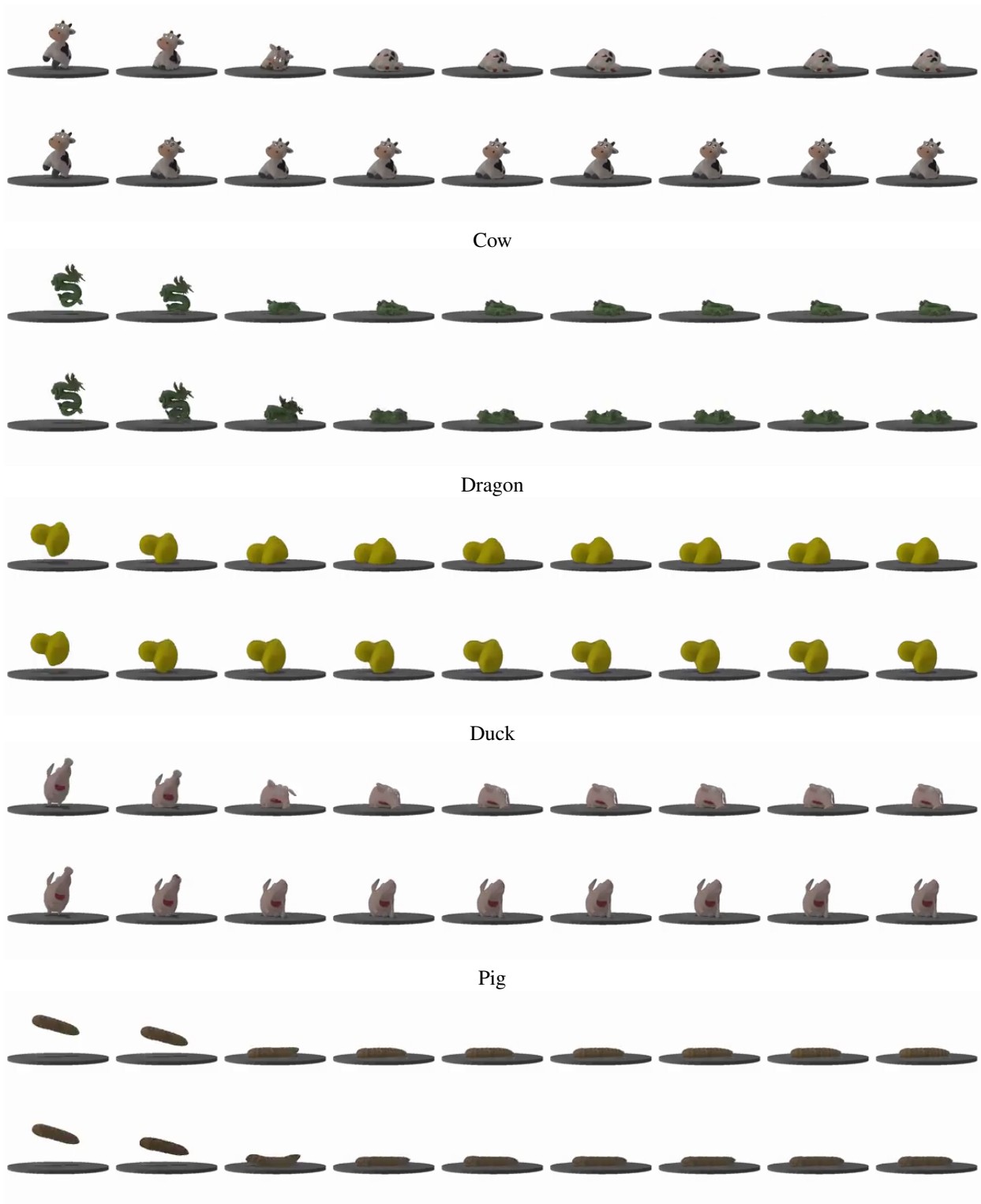

*Figure S19.* CosmosFT: Various elastic simulations for different objects.

### F.4.3. RIGID BODY DYNAMICS

Finally, (Fig. S20 - S21) showcase rigid body collisions involving objects with complex shapes (e.g., Squirrel, Turtle). Here, the primary challenge is maintaining geometric rigidity upon impact.

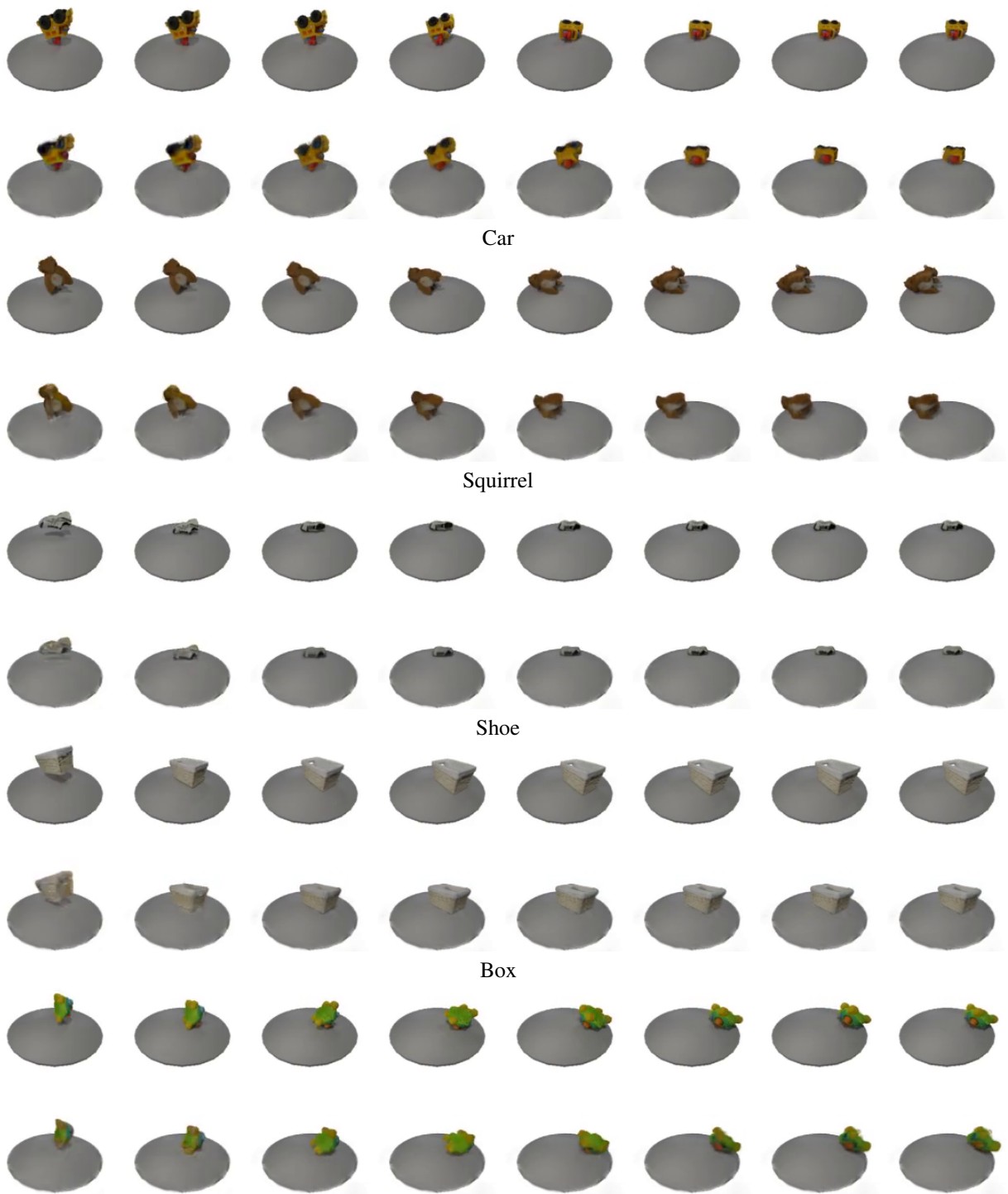

Car

Squirrel

Shoe

Box

Turtle

*Figure S20.* 3DGSim: Various rigid simulations for different objects.

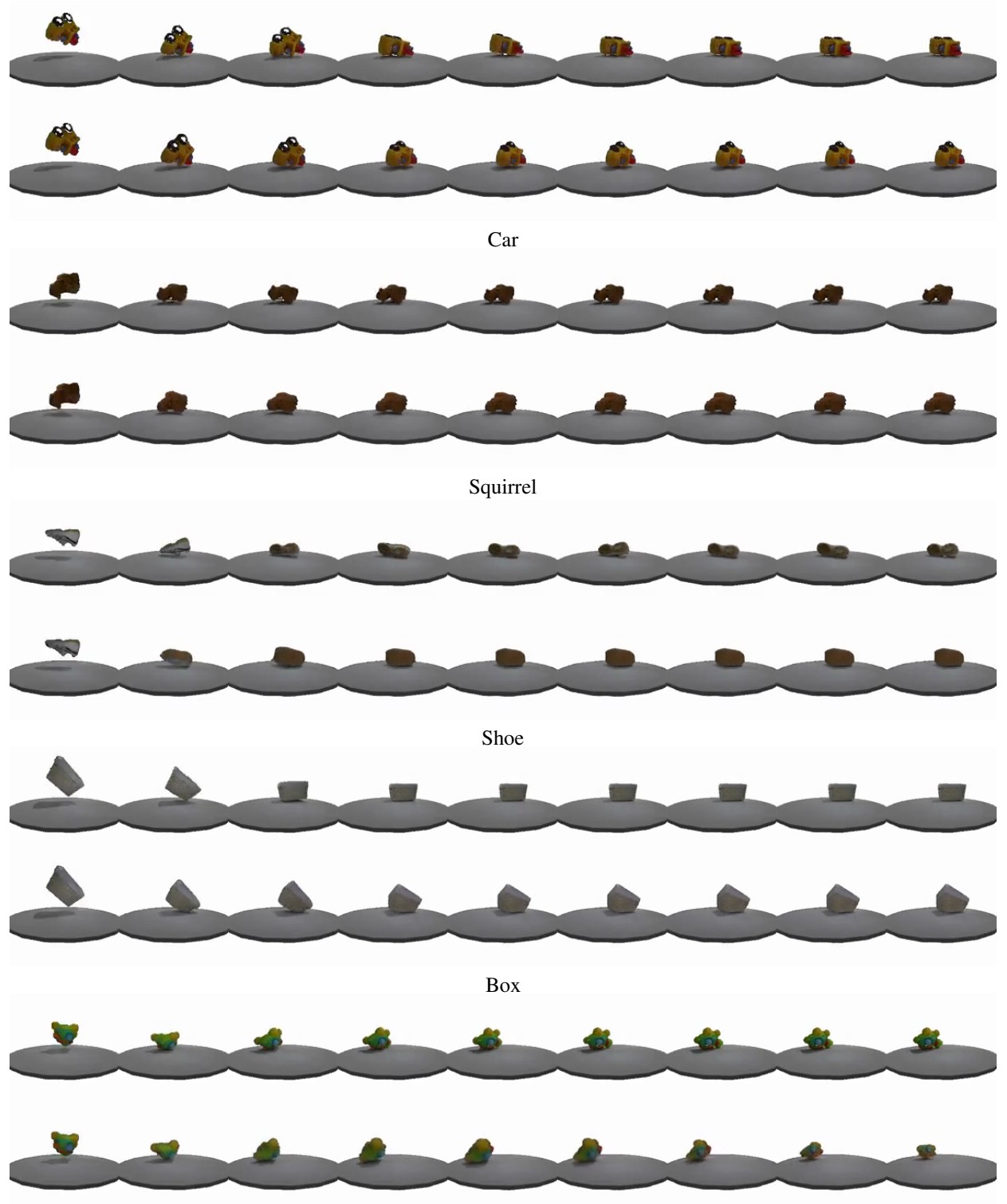

Car

Squirrel

Shoe

Box

Turtle

*Figure S21.* CosmosFT: Various rigid simulations for different objects.

