# OpenReview forum: "Learning 3D-Gaussian Simulators from RGB Videos"
_ICML.cc/2026/Conference — ICML 2026 regular_

### Official Review · Reviewer_n6s6 · 2026-02-14

**Soundness:** 2
**Presentation:** 2
**Significance:** 1
**Originality:** 3
**Overall Recommendation:** 3
**Confidence:** 4

**Summary:**

This paper introduces 3DGSim, a fully end-to-end and differentiable framework for 3D physical simulation directly from multi-view RGB videos. The method integrates a feed-forward inverse renderer based on MVSplat to reconstruct 3D Gaussian, a Transformer-only dynamics model for efficient spatiotemporal reasoning, and a 3DGS renderer for image-level supervision. 3DGSim support learning latent visuo-physical particle representations that jointly capture geometry, appearance, and dynamics without relying on explicit physics priors. Experiments demonstrate physically plausible long-horizon predictions and consistent novel-view synthesis across rigid, elastoplastic, and cloth-like scenes.

**Compliance With Llm Reviewing Policy:**

Affirmed.

**Final Justification:**

During the rebuttal period, the authors often used differences in paradigms (e.g., requiring manual intervention or GPT hints) as reasons to avoid providing additional comparisons, until another **reviewer ziXr**, raised a similar concern, after which they added the comparison with [1].

However, my concern regarding the limited physical simulation in collision and object interaction still remains. While the additional comparison with [1] alleviates my concern about insufficient evaluation to some extent, it does not fully resolve it.

In summary, I decide to change my score to Weak Reject.


[1] Zhong, L., Yu, H. X., Wu, J., & Li, Y. (2024). Reconstruction and simulation of elastic objects with spring-mass 3d gaussians. In European Conference on Computer Vision (pp. 407-423). Cham: Springer Nature Switzerland.

**Key Questions For Authors:**

1. The authors claim open-source release as one of their main contributions; however, no code or dataset is included in the supplementary material. How this can be considered a main contribution of the paper.

2. All evaluations are limited to synthetic objects. It is unclear if the proposed method can be effectively applied to real-world scene.

**Limitations:**

yes

**Strengths And Weaknesses:**

Strengths:

1. The paper cleverly uses PTv3 to avoid the reliance on KNN graph construction, which can significantly improve inference speed.

2. The authors provide extensive experiments in the main paper, supplementary material, and videos across different materials, validating the generalization and effectiveness of the method.

Weaknesses:
1. The limited comparison and evaluation is my main concern. Only Cosmos and fine-tuned Cosmos are compared. In Lines 308–310, the authors mention that “DPI-Net and VGPLD are open-source but rely on ground-truth particle trajectories and require major adaptation to fit our setting.” However, this cannot be considered a scientific reason for avoiding comparisons. Moreover, there are still many available GS-based physical simulation methods that could be used as baselines, including but not limited to PhysGaussian [1], PhysDreamer [2], and Spring-Mass 3D Gaussians [3]. I suggest the authors conduct a more careful review and include additional baselines to strengthen the validation of the proposed method.

2. Current SOTA methods such as PhysGen3D [4] support user-interactive physical simulation (e.g., defining the motion direction of objects). In contrast, the videos and demos shown in this paper suggest that the proposed method only supports simple falling and top-down collision scenarios, which limits its applicable use cases.

3. The paper lacks some important details, including information about the training datasets, inference time, as well as the visualization and construction procedures for the introduced rigid-body, elastic, and cloth datasets.

4. No physics-related metrics are used for evaluation. If the core of the paper is kinematic simulation and interpolation of intermediate processes, PSNR alone is insufficient to assess physical simulation performance. I suggest adding metrics such as motion realism, motion smoothness, or other physics-aware criteria to further evaluate the proposed approach.

5. The paper lacks a workflow figure that provides a general overview of the simulation pipeline.


[1] Xie, T., Zong, Z., Qiu, Y., Li, X., Feng, Y., Yang, Y., & Jiang, C. (2024). Physgaussian: Physics-integrated 3d gaussians for generative dynamics. In Proceedings of the IEEE/CVF Conference on Computer Vision and Pattern Recognition

[2] Zhong, L., Yu, H. X., Wu, J., & Li, Y. (2024). Reconstruction and simulation of elastic objects with spring-mass 3d gaussians. In European Conference on Computer Vision (pp. 407-423). Cham: Springer Nature Switzerland.

[3] Zhang, T., Yu, H. X., Wu, R., Feng, B. Y., Zheng, C., Snavely, N., ... & Freeman, W. T. (2024). Physdreamer: Physics-based interaction with 3d objects via video generation. In European Conference on Computer Vision (pp. 388-406). Cham: Springer Nature Switzerland.

[4] Chen, B., Jiang, H., Liu, S., Gupta, S., Li, Y., Zhao, H., & Wang, S. (2025). Physgen3d: Crafting a miniature interactive world from a single image. In Proceedings of the Computer Vision and Pattern Recognition Conference (pp. 6178-6189).

---

> ### Author Rebuttal · Authors · 2026-03-30
>
> We thank the reviewer for their time invested in reviewing our work. We agree that the comparison to SOTA baselines is an integral part of scientific work. However, the methods listed by the authors solve only small specific subsets of the problem setting that we address in our work.
>
> Our method - 3DGSim - learns to predict and render physics simulation using solely multi-view videos. This problem has been so far only tackled by video generation models such as Cosmos, which we do use as baseline.
>
> **# W1a: DPI-Net and VGPL**
>
> As noted above, DPI-Net and VGPL assume known particle trajectories during training, whereas 3DGSim is trained purely from multi-view video. They operate on the order of ~100 particles, while our method handles ~50K, and they only simulate simple amorphous shapes or point masses, whereas our experiments include complex objects such as the Stanford dragon. Moreover, these methods rely on heuristically encoded scene boundary information in the GNN. In effect, the reviewer is asking us to run these methods on simulations that were not reported even in their original papers, likely due to computational constraints.
>
> A direct comparison is therefore neither practical nor informative: our method reconstructs and simulates the entire scene as perceived in the input images, including boundaries, while DPI-Net and VGPL assume perfect mesh knowledge throughout all time steps and use constraint heuristics for the rest of the scene.
>
> **# W1b: GS-based Simulation Baselines**
>
> The methods the reviewer mentions [1-4] occupy a fundamentally different category: they all rely on black-box analytical simulators (typically MPM or Spring-based) to produce dynamics. The main difference is in how the simulation parameters are obtained: [1] uses hand-tuning, [2] and [3] use per-scene optimization, and [4] uses a language model. In contrast, 3DGSim learns dynamics directly from videos in a fully feed-forward, end-to-end manner, requiring no physical priors, no manual boundary conditions, and no per-scene optimization.
>
> We discuss these distinctions in detail in Appendix C of the supplementary and will gladly expand that section to include [2,4] for completeness.
>
> **# W2: User-Interactivness**
>
> We would like to draw the reviewer's attention to Section 5.4 and Experiments E4–E6 of the supplementary material, where we demonstrate interaction with scenes via sparse hand points (21 per hand). There, we show that 3DGSim can model one-arm and two-arm movements and their effects on the rest of the scene, including, for example, object handovers between hands.
>
> **# W3: Datasets & Inference Details**
>
> The dataset details (construction, simulation, sizes) are discussed in the third paragraph of Section 5 in the main paper. Inference time is covered in Section D of the supplementary material. If there are specific details the reviewer feels are still missing, we would be happy to add them.
>
> **# W4: Other Metrics**
>
> Thank you for the suggestion, in the revised manuscript we report FVD to capture spatio-temporal coherence.
>
> We have added a comprehensive overview of all evaluation metrics to the supmat of the camera-ready version. Below we show the main results with the addition of FVD:
>
>
> | Dataset | Model | PSNR (future) ↑ | SSIM ↑ | LPIPS ↓ | FVD ↓ |
> |--|--|--|--|--|--|
> | Elastic | 3DGSim 4-12 latent | 33.15| 0.97 | 0.02 | 91.3 |
> | - | 3DGSim 4-12 explicit | 29.69 | 0.92 | 0.09 | 215.0 |
> | - | CosmosFT | 26.97 | 0.82 | 0.07 | 156.4 |
> | Cloth | 3DGSim 4-8 latent | 26.46 |0.88 | 0.09 | 117.5 |
> | - | 3DGSim 4-8 explicit | 24.40 | 0.86 | 0.12 | 112.7 |
> | - | CosmosFT | 23.12 | 0.72 | 0.15 | 134.5 |
> | Rigid | 3DGSim 4-12 latent | 28.35 | 0.91 | 0.08 | 297.4 |
> | - | 3DGSim 4-12 explicit | 25.50 | 0.85 | 0.17 | 372.6 |
> | - | CosmosFT | 27.31 | 0.70 | 0.11 | 351.5 |
>
> Please note that **state-level metrics (e.g., particle position error) are not applicable in our setting**, since 3DGSim learns its particle representation purely from images there is no correspondence to the simulator's internal state.
>
> **# W5: Workflow / Pipeline Figure**
>
> We intended Figure 1 to serve as the pipeline overview: it shows how input images are consumed by the inverse renderer (MVSplat or DepthAnything3) to reconstruct latent particle features, which are then processed by the dynamics model over an input window of 4 timesteps to predict the future auto-regressively. Image reconstruction loss is used for training (or finetuning in case of DA3) all modules end-to-end.
>
> **# Q1: Release**
>
> We are committed to releasing our code and datasets and will do so for the camera-ready version.
>
> **# Q2: Real-World Applicability**
>
> We kindly ask the reviewer to consider the real-world experiments in Section 5.4 of the main paper and E4-6 of the supplementary in their assessment of our work. In these experiments, **our proposed model successfully simulates real-world human arms manipulating real-world household objects**.

---

> > ### Author Rebuttal · Reviewer_n6s6 · 2026-03-31
> >
> > Thanks for your reply, however I still think some of my concerns have not been fully addressed:
> >
> > 1. For answer #W1b, I do not fully agree with the argument presented in the appendix that per-scene optimization methods can be excluded from comparison simply because your approach is end-to-end. E.g. [1], objective comparsion would still be necessary if it has better performance.
> >
> > 2. For answer #W2 and #Q2 The provided real world demos seem just have simple interactions between hands and single objects, lack more complex object interactions, such as object exchange, rich collisions, complex motions, or deformation/crushing behaviors. I think my concern regarding the insufficiency of the evaluated scenarios still unresolved.
> >
> > [1] Zhong, L., Yu, H. X., Wu, J., & Li, Y. (2024). Reconstruction and simulation of elastic objects with spring-mass 3d gaussians. In European Conference on Computer Vision (pp. 407-423). Cham: Springer Nature Switzerland.

---

> > > ### Author Response · Authors · 2026-04-02
> > >
> > > We thank the reviewer for the timely reply.
> > >
> > > 1. We want to respectfully point out that the reviewer’s disagreement does not yet engage with the main technical obstacles to a fair comparison between SpringGaus [1] and 3DGSim, which we have outlined in both the supplementary material and the rebuttal. We would therefore like to restate these points more clearly.
> > >
> > > - **First, [1] assumes that scene constraints are known a priori:** boundary constraints, gravity, plane normals/positions, and damping are hand-specified before learning begins. In contrast, 3DGSim infers such constraints directly from images during training. This difference is fundamental, because it gives [1] access to information that our method is explicitly designed to recover on its own.
> > >
> > > - **Second, [1] is restricted to ideal elastic deformations** and does not model cloth or rigid-body simulations. By contrast, the deformations considered in our experiments are elasto-plastic, rigid, and simulate cloth. As a result, [1] must be substantially extended before it could be applied to the physical regimes studied in our work. Without such extensions, a comparison would not only be difficult to interpret, but also misaligned with the actual scope of the two methods.
> > >
> > > - **Third, [1] relies on multi-stage optimization on a single specific trajectory.** This raises a practical concern for any attempted comparison: would one reconstruct the object and tune parameters on each individual test trajectory? In contrast, 3DGSim learns purely from image-based training data in an end-to-end manner and only after training the model is used on the test data.
> > >
> > > Please also consider these additional practical problems that hinder a comparison to the other per-scene optimization methods:
> > >
> > > - **PhysDreamer does not model contacts**.
> > > - PhysGaussian reconstructs static 3D scenes from multi-view images, separates the object, manually specifies ground boundary conditions, and then simulates motion using a hand-tuned MPM simulator. **Physics are not learned**.
> > > - PhysGen3D prompts GPT-4o to suggest physical parameters (density, elasticity) and uses these in an MPM simulator. **Physics are not learned**.
> > >
> > > For these reasons, our concern is not a reluctance to compare against [1], but rather that **there exist substantial technical issues that currently prevent such a comparison from being informative and methodologically sound.**
> > >
> > > 2. Our model is the **first particle-based simulator that is able to predict real-world object-to-human hand-overs**. This is accomplished **using only ~60 training trajectories**. Clearly, we would also like to train the model on more data that shows even more complex real-world physics, but unfortunately available data sets, such as the HO-CAP dataset, are quite restricted in size. We therefore respectfully ask the reviewer to assess our contribution in the context of the state of the art in particle-based simulation, where the ability to predict real-world object-to-human hand-overs from only 54 training trajectories constitutes a significant step forward.
> > >
> > > **Edit:**  Since Reviewer ziXr also requested a comparison with [1], we have taken the time to run Spring-Gaus on our benchmark. The full results and discussion can be found in our [response to Reviewer ziXr](https://openreview.net/forum?id=Icj19tWHI1&noteId=ykffvpVXzC). We remain of the opinion that this comparison is not fully fair due to the substantial differences in problem setup outlined above: Spring-Gaus requires ground-truth scene constraints, optimizes and tests on the same trajectory, cannot handle cloth or particle-collisions in general, and does not simulate visual effects. Nonetheless, the results show that 3DGSim outperforms Spring-Gaus on future prediction across all metrics despite requiring no privileged inputs or per-trajectory optimization. We hope this additional experiment addresses the reviewer's remaining concern regarding baseline comparisons.

---

### Official Review · Reviewer_ziXr · 2026-02-24

**Soundness:** 3
**Presentation:** 2
**Significance:** 2
**Originality:** 3
**Overall Recommendation:** 4
**Confidence:** 4

**Summary:**

This paper presents 3DGSim, a differentiable framework for predicting 3D particle dynamics from multi-view RGB videos. The architecture combines an MVSplat-based inverse renderer for view-independent feature extraction with a transformer-only dynamics model that includes a Temporal Encoding and Merging module, along with a Gaussian Splatting head for efficient rendering-based supervision. Experimental evaluations on datasets involving rigid objects, elastic and cloth objects demonstrate that the model accurately predicts complex interactions and can generalize to scenarios that differ from the training data. Additionally, the framework shows practical potential through successful preliminary applications in action-conditioned object manipulation.

**Compliance With Llm Reviewing Policy:**

Affirmed.

**Final Justification:**

After reviewing the authors' responses, I found that many of my concerns were addressed. However, the paper still has significant limitations. For example, the vision-only model does not fully capture physical states and cannot be directly applied to real-world objects, which often vary greatly in shape, texture, and material properties. These limitations may hinder the practical effectiveness of using the proposed method for understanding physics in real-world scenarios or for robotics tasks. That said, the overall pipeline could be useful for future research to build upon. Therefore, I will maintain my initial recommendation.

**Key Questions For Authors:**

1.	It appears that the particle’s position is derived from the reconstructed 3D Gaussian splats, which are primarily distributed around the object's surface. How does the proposed method address Continuum Mechanics when physical particles are only located at the surface?
2.	The authors should clarify whether there are specific requirements for the input sequence of past states. It is unclear how the dynamics model can differentiate between objects with identical geometries but different material properties (e.g., rigid vs. elastic) if the provided history only covers a short free-fall period. Since the trajectories and rendered appearances are nearly identical prior to collision, what mechanism allows the model to infer these distinct physical behaviors?
3.	Is it able to input a varied number of past frames as the input to the dynamics model once it is trained (e.g., training with 4 past frames and testing with 2 or 8 past frames)? If so, how does the variation during inference time influence the performance for dynamics prediction?
4.	The authors state in Section 5 that there are few open-source 3D baselines and datasets available for comparison. However, several studies [a-c] that focused on learning physical dynamics from multi-view videos have released their resources. It is strongly recommended that the authors discuss and compare their work with these existing baselines in the manuscript.
5.	How does the proposed method perform when: (a) encountering unseen object shapes; (b) handling interaction between rigid and elastic objects?
6.	The authors should clarify the rationale for solely relying on rendering quality as the evaluation metric for a physics simulator. Given that the primary experiments utilize simulated data, incorporating the Chamfer distance between predicted and ground-truth particle trajectories would provide a more meaningful measure of physical accuracy.

[a] Reconstruction and Simulation of Elastic Objects with Spring-Mass 3D Gaussians. ECCV 2024.

[b] NeuMA: Neural Material Adaptor for Visual Grounding of Intrinsic Dynamics. NeurIPS 2024.

[c] PhysTwin: Physics-Informed Reconstruction and Simulation of Deformable Objects from Videos. ICCV 2025.

**Limitations:**

- The method requires multi-view, calibrated RGB videos to learn object dynamics, which is relatively hard to set up in real-world scenarios.

- The paper primarily considers objects with uniform textures, which may indicate a limitation in its ability to learn dynamics for objects with complex textures.

**Strengths And Weaknesses:**

**Strengths**
+ This paper presents a novel feed-forward particle dynamics model that can predict physically plausible dynamics for simple geometries.
+ The particle dynamics model achieves near real-time simulation speeds, which can accelerate robot learning in simulation environments.

**Weaknesses**
- The experiments only evaluated a limited number of baselines, and some related work was neither discussed nor compared.
- The datasets used in this paper only cover relatively simple shapes and mainly uniform textures.

---

> ### Author Rebuttal · Authors · 2026-03-30
>
> ## Simple shapes and textures
>
> Compared to prior works (DPI-Net, VGPLD, VPD, DEL ..), we intentionally chose objects with non-uniform textures, complex geometries (e.g. Stanford Dragon), and real-world scenarios to raise the visual bar. That said, we agree that real-world objects exhibit higher variance in shape, texture, and material properties. Scaling to this level of diversity remains an important next step.
>
> ## 1. Continuum mechanics without volumetric particles
>
> Indeed, continuum mechanics simulators such as FEM and MPM discretize volume instead of surface geometry. These methods require a volume discretization as analytical laws of material deformation require  **small elements** to yield accurate predictions. In comparison, a surface discretization requires a model to learn the effect of all internal material deformations onto the boundary elements (taking external/constraint/inertial forces into account). This is a **highly nonlinear function that depends on all boundary elements**, but given enough data, our experiments show that a neural network can learn this function.
>
> ## 2. How the model infers physical properties
>
> In our interpretation, the model **combines geometry and visual cues to infer an object's most probable physical properties** (stored in latent features) and uses the **past window to capture dynamic state**. The model would fail on strongly OOD cases (e.g. a tennis ball filled with tungsten), but given a sufficiently diverse training dataset, we expect it to reproduce the physics of most real-world household objects, enabling its use in training universal robot control policies.
>
> ## 3. Variable-length input at inference time
>
> Thank you for the interesting question. We evaluated the elastic dynamics as suggested. Performance peaks at 4 past frames as this matches the training configuration. With fewer steps (1–2), the model lacks sufficient temporal context. With more steps (8), performance degrades because TEM only learns positional embeddings for the first two hierarchy levels (matching 4 input steps); deeper levels are engaged with untrained embeddings, corrupting feature representations.  That said, even in these cases the dynamics **remain qualitatively plausible**, with degradation appearing mainly as reduced post-collision accuracy. Training with variable-length input windows is an interesting direction for future work.
>
> | past frames | PSNR (future) ↑ | SSIM ↑ | LPIPS ↓ | FVD ↓ |
> |:---:|:---:|:---:|:---:|:---:|
> | **4** | **33.15** | **0.97** | **0.02** | **91.3** |
> | 8 | 28.60 | 0.95 | 0.04 | 242.2 |
> | 2 | 27.27 | 0.92 | 0.08 | 387.8 |
> | 1 | 22.92 | 0.87 | 0.14 | 959.2 |
>
>
> ## 4. Comparison with  3D baselines
>
> SpringGauss, NeuMA, PhysTwin and similar methods rely on per-scene iterative optimization with hand-engineered components, anallytical backend simulators (MPM, spring-mass), material-specific parameters (Young's modulus, stiffness, etc), manually defined boundary conditions, and privileged inputs (RGBD, segmentation, 3D tracking). They reconstruct only dynamic objects (not full scenes) through separate geometry, physics, and appearance stages, requiring a new optimization for every object. In contrast, 3DGSim learns dynamics directly from videos in a fully feed-forward, end-to-end manner, requiring no physical priors, no manual boundary conditions, and no per-scene optimization.  We will also discuss these methods in the supmat.
>
> ## 5. Combinatorial generalization to unseen objects/materials
>
> Despite training only on single-object ground collisions, 3DGSim generalizes to unseen shapes (red/blue bunny in Figs. S6–S9, though sensitive to color), multi-body dynamics (up to 5 objects), and modified scenes (ground removal/height changes), as shown in Figs. 9, 10, 12. Moreover, our real-world experiments (Sec. 5.4) demonstrate soft-rigid interaction through hand-object manipulation scenarios.
>
> ## 6. Rationale for image-based evaluation
>
> The particles of 3DGSim's and the simulator are fundamentally different: they differ in number, spatial distribution, coverage (ours include the full scene vs. the simulator tracks only dynamic bodies), and semantics (appearance/latent features vs. physical state). The reconstruction is non-unique, so positional differences do not necessarily indicate dynamics errors: post-hoc registration (e.g. nearest-neighbor matching) would conflate reconstruction with dynamics accuracy. This separation is by design: 3DGSim learns to simulate without GT physical state, keeping it applicable to real-world scenarios. **Evaluating against that state penalizes the method for the very property that makes it general.**
>
> That said, since MVSplat and DA3 are pixel-aligned depth models (constrained to surfaces), whether multi-layer depth models [2,3] with dynamics-based regularization could recover volumetric structure is an interesting open question.
>
> [2] R. Tucker et al. Single-view view synthesis with multiplane images
>
> [3] Z. Li et al. Amodal depth anything

---

> > ### Author Rebuttal · Reviewer_ziXr · 2026-04-02
> >
> > I appreciate the feedback from the authors. However, several of my initial concerns remain after reading the authors' feedback.
> >
> > 1. The authors assert that "given enough data, our experiments show that a neural network can learn this function." However, because the training data is sourced exclusively from RGB videos, the model can only observe visual surfaces. Standard cameras cannot capture important physical attributes such as mass distribution, internal stress, stiffness, or volumetric density. Therefore, I question whether the model could accurately simulate plausible dynamics for two visually opaque objects—one hollow and the other solid—based solely on a few initial frames.
> > 2. The authors do not directly address my second question: "How can the dynamics model differentiate between objects with identical geometries but different material properties (e.g., rigid vs elastic) if the provided data only covers a short free-fall period?" While the authors assert that "given a sufficiently diverse training dataset, we expect it to reproduce the physics of most real-world household objects," it remains unclear how diverse this training dataset needs to be to achieve that capability.
> > 3. While I recognize the methodological differences between feed-forward architectures and per-scene optimization baselines, the ultimate task—simulating the dynamics of 3D objects from video—remains constant. A purely textual discussion of these differences is insufficient to capture their implications. The authors highlight that traditional baselines require privileged inputs, such as RGB-D data and segmentation masks, whereas 3DGSim relies solely on RGB input. This characteristic should be viewed as a strength of the proposed method rather than a justification for neglecting quantitative comparisons. However, it is important to note that baselines benefit from per-scene optimization, which, when combined with analytical solvers, ensures physical consistency within the target scene. In contrast, 3DGSim employs a learned feed-forward pass, which raises questions about its performance relative to those baselines. Without direct comparisons, it remains unclear how effectively 3DGSim captures the physical dynamics that the baselines guarantee.

---

> > > ### Author Response · Authors · 2026-04-06
> > >
> > > ## Q1 & Q2: Limits of vision-only physical inference
> > > We thank the reviewer for the follow-up. We believe both questions address the same underlying concern: how can a vision-only model disambiguate objects that appear identical in the input but behave differently physically? This applies both to objects with differing internal structure (hollow vs solid, Q1) and to objects with differing material properties observed during a phase where their dynamics are indistinguishable (rigid vs elastic in free-fall, Q2).
> > >
> > > Our model can only infer physical properties from three sources of information: geometry, visual appearance, and temporal context (observed motion within the input window, including post-contact behavior when available). For the vast majority of real-world objects, these signals are sufficient. A rubber ball and a steel ball, for instance, differ in color, texture, and post-contact motion. However, when two objects are visually and geometrically identical yet physically distinct (e.g., a hollow vs solid sphere of the same material, or rigid vs elastic objects observed only before contact), no vision-only method can disambiguate them without additional interaction or observation. This includes human perception. We acknowledge this as a fundamental limitation of any vision-only approach, not specific to our method. How to best resolve such ambiguities is an open question that we find equally interesting. One possible direction could be post-observation inference-time optimization of the *static latent features*, but we leave this for future investigation.
> > >
> > > ## Spring-Gaus Comparison
> > >
> > > We evaluated Spring-Gaus with default parameters on 3 test-set trajectories per object, providing ground-truth ground height and object masks as required inputs. For velocity-estimation we use the first 4 frames. Several caveats apply to this comparison.
> > >
> > > First, Spring-Gaus only supports contact via a boundary condition against a known ground plane and does not handle general particle collisions, ruling out cloth dynamics and multi-object interaction. Second, its estimated parameters (per-anchor stiffness, rest lengths, connectivity, ..) are tied to a specific point cloud topology derived from random sampling of the static reconstruction. Since there is no correspondence between anchor points across different trajectories, parameters cannot transfer, forcing optimization and testing on the same trajectory (length 84 for elastic, 64 for rigid).  Third, the method does not simulate visual effects such as shadows or lighting, which 3DGSim handles through its learned implicit simulator.
> > >
> > > Results reflect these constraints. Soft dynamics are better estimated, though plasticity remains uncaptured. Rigid bodies recover appropriately stiff springs, but bounciness leads to poor overall accuracy. In the full-scene setting, the lack of visual simulation accounts for an additional 2-3 dB PSNR reduction.
> > >
> > > | Dataset     | Model              | PSNR (future) ↑  | PSNR (past) ↑ | SSIM (future) ↑ | LPIPS (future) ↓ |
> > > | :---------- | :----------------- | :--------------- | :------------ | :-------------- | :--------------- |
> > > | **Elastic** | 3DGSim 4-12 latent | **33.15 ± 3.51** | 34.42 ± 2.34  | **0.97 ± 0.02** | **0.02 ± 0.01**  |
> > > |             | Spring-Gaus  †      | 24.13 ± 3.65     | 37.20 ± 2.08  | 0.94 ± 0.02     | 0.08 ± 0.02      |
> > > |             | Spring-Gaus (obj only)        | 27.13 ± 3.33 | 37.20 ± 2.08  | 0.97 ± 0.01 | 0.04 ± 0.01 |
> > > | **Cloth**   | 3DGSim 4-8 latent  | **26.46 ± 2.62** | 34.89 ± 2.23  | **0.88 ± 0.03** | **0.09 ± 0.02**  |
> > > |             | Spring-Gaus        | --               | --            | --              | --               |
> > > | **Rigid**   | 3DGSim 4-12 latent | **28.35 ± 2.70** | 32.86 ± 1.51  | 0.91 ± 0.03 | **0.08 ± 0.02**  |
> > > |             | Spring-Gaus †       | 23.85 ± 1.01     | 33.89 ± 1.39  | 0.89 ± 0.02     | 0.13 ± 0.03      |
> > > |             | Spring-Gaus (obj only)        | 27.39 ± 1.47 | 35.81 ± 1.68 | **0.94 ± 0.01** | 0.08 ± 0.02 |
> > >
> > > ** Spring-Gaus † *includes a static ground reconstruction for rendering; the lower scores reflect uncaptured visual effects (shadows, lighting).*
> > >
> > > ** *Spring-Gaus cannot handle cloth dynamics due to lack of general collision support.*
> > >
> > >
> > > **Edit:** Added object-only metrics evaluating predictions within the object mask only.
> > >
> > >
> > > ---
> > >
> > > We hope that this clarification together with the additional experiments addresses the reviewer's remaining questions.

---

### Official Review · Reviewer_chRQ · 2026-03-11

**Soundness:** 3
**Presentation:** 3
**Significance:** 3
**Originality:** 3
**Overall Recommendation:** 4
**Confidence:** 4

**Summary:**

This paper tackles the task of physical simulation based on 3D Gaussians and learning from videos. For this, it develops a system that 1) lifts multiview images to 3DGS with features through MVSplat; 2) predicts dynamics for each particle in the 3DGS set. Specifically, they predict the 3D position as well as feature change. The authors end-to-end train the whole system with photometric reconstruction loss. Experiments on various datasets demonstrate the effectiveness of the proposed approach.

**Compliance With Llm Reviewing Policy:**

Affirmed.

**Final Justification:**

My concerns are mostly addressed, and I am looking forward to the promised results in the camera-ready. Even considering only the current version, I intend to maintain my score.

**Key Questions For Authors:**

See "weakness".

**Limitations:**

Not discussed enough.

**Strengths And Weaknesses:**

## Strengths

- soundness: the experiments and ablations demonstrate the effectiveness of the approach;
- presentation: the paper is well-written and easy to follow;
- significance: the task of learning dynamics from videos is vital for downstream tasks, e.g., AR/VR and robotics;
- originality: the design of various modules, e.g., temporal merging.

## Weakness

### 1. Possibility of a unified model

If I understand correctly, the current model is trained separately on different datasets. I am curious to know how much worse the performance will be if we use the same budget but directly train on the combined datasets.

### 2. More ablations

I would like to understand how important the following is:
- Eq. (4): What if we unify the feature and do not separate the static and dynamic parts
- Eq. (5): What if we directly predict positions and dynamic features instead of predicting the deltas, since this could potentially recover from suboptimal initial positions or features?

---

> ### Author Rebuttal · Authors · 2026-03-30
>
> We thank the reviewer for the questions and time invested that helped us to further polish our work.
>
> ### 1. Possibility of a unified model
>
> > If I understand correctly, the current model is trained separately on different datasets. I am curious to know how much worse the performance will be if we use the same budget but directly train on the combined datasets.
>
> This would indeed be an interesting experiment to run. As training new experiments takes longer than the span of the rebuttal period, we are currently running an experiment where we learn such a unified model on cloth and elastic datasets and will report the results in the camera-ready version.
>
> ### 2. More ablations
>
> I would like to understand how important the following is:
>
> > Eq. (4): What if we unify the feature and do not separate the static and dynamic parts
>
> Our representation evolved through systematic exploration:
> We initially used **fully-static explicit 3DGS features**, updating only positions. This worked but struggled with shadows, which had to be approximated solely through positional changes. Moving to **fully-dynamic explicit features** resolved this, and adding a static latent feature fed into the dynamic model further improved results and training speed. However, operating in explicit space required rotations to respect SO(3) constraints and scale updates to be tightly controlled, making training harder to stabilize.
> Our **final design**, static & dynamic latent features, eliminates these constraints and achieves the best performance, as confirmed by our ablation (Table S1 - in the camera ready version).
> We acknowledge that a **fully-dynamic latent** variant (without the static split) is an interesting additional ablation and we will include this comparison in the camera-ready version.
>
>
>
> > Eq. (5): What if we directly predict positions and dynamic features instead of predicting the deltas, since this could potentially recover from suboptimal initial positions or features?
>
> Thank you for this suggestion. For features, we actually do predict the value directly rather than a delta, which performs slightly better (we use the delta notation for simplicity, as PTv3 follows a residual-style architecture). For positions, however, predicting absolute coordinates introduces two key challenges: (1) the large scale of absolute xyz values makes training less stable, particularly early on when the dynamic model may try to compensate for encoder errors, and (2) we lose the spatial invariance that comes with local, relative predictions. The delta formulation naturally avoids both issues.

---

> > ### Author Rebuttal · Reviewer_chRQ · 2026-04-05
> >
> > I thank the authors for their time and effort in addressing my concerns.
> >
> > I think the clarifications are good, and I am looking forward to those promised results in the camera-ready.
> >
> > Overall, I am inclined to maintain my current score.

---

### Official Review · Reviewer_t9ck · 2026-03-13

**Soundness:** 3
**Presentation:** 3
**Significance:** 3
**Originality:** 3
**Overall Recommendation:** 4
**Confidence:** 4

**Summary:**

**Task Description**
- This paper addresses the problem of learning 3D physical simulation directly from multi-view RGB videos, where the goal is to predict future scene states given a short history of multi-view observations, without requiring ground-truth particle trajectories, depth supervision, or explicit physics priors.

**Research Motivation**
- Existing particle-based simulators rely on GNN-based graph construction with kNN, which is computationally expensive and requires hand-crafted edge features, limiting scalability. Meanwhile, 2D video generation models lack 3D structure awareness, leading to failures in spatial consistency and object permanence. This paper is motivated by the question of whether the inductive bias from locally connected graphs can be replaced by scalable transformer-based computation while still learning physically grounded 3D simulation from raw RGB observations.

**Contributions**
- The paper introduces 3DGSim, a fully differentiable framework consisting of three jointly trained components: (i) a feed-forward inverse renderer extending MVSplat with a FiLM-conditioned feature encoding network that produces view-independent latent particle representations, (ii) a transformer-based dynamics model extending PTv3 with Temporal Encoding and Merging (TEM) that replaces kNN graph construction with space-filling curve serialization for spatiotemporal reasoning, and (iii) a Gaussian Splatting decoder that renders predicted particle states into images for end-to-end supervision via reconstruction loss.

---

**Compliance With Llm Reviewing Policy:**

Affirmed.

**Final Justification:**

3DGSim presents a fully differentiable framework for learning 3D physical simulation from multi-view RGB videos, replacing GNN-based kNN graph construction with PTv3's space-filling curve serialization and training end-to-end with image reconstruction loss alone—without ground-truth trajectories, depth supervision, or explicit physics priors.

- **Soundness.** The evaluation demonstrates consistent improvements across three datasets, and the rebuttal adequately justified why trajectory-level metrics are ill-defined in this setting. Quantitative evaluation remains limited to synthetic scenes, but this is standard for vision-based learned simulators.

- **Originality.** The temporal encoding and merging mechanism extending space-filling curve serialization to the spatiotemporal domain is an elegant design. The contribution lies primarily in integrating existing components (MVSplat, PTv3, 3DGS) into a coherent end-to-end framework, but the combination is well-motivated and non-trivial.

- **Presentation.** The paper is well-written with clear positioning and effective visualizations of the out-of-distribution generalization capabilities.

- **Significance.** The compelling OOD generalization results provide strong evidence that the representation captures spatially grounded physics. The evaluation scope limitation tempers the impact, though the rebuttal reasonably explains this as a constraint of the vision-only paradigm.

**Overall assessment.** The rebuttal addressed all of my concerns in a well-reasoned manner. While inherent evaluation limitations remain—synthetic-only quantitative evaluation and the absence of trajectory-level physics metrics—these are well-justified as fundamental constraints of the vision-only simulation paradigm shared with prior work, rather than oversights of this specific paper. The core contributions (end-to-end differentiable pipeline, space-filling curve serialization for spatiotemporal dynamics, compelling OOD generalization) are solid and likely to be built upon. I am maintaining my score of **4 (Weak Accept)**.

**Key Questions For Authors:**

### Key Questions for Authors
- (Related to W2) Can the paper provide trajectory-level physical accuracy metrics (e.g., particle position error against ground-truth Genesis simulator states, or energy/momentum conservation measures) in addition to image-based metrics? Since ground-truth particle states are available from the simulator, this should be straightforward to compute and would separate dynamics accuracy from rendering quality.
- (Related to W3) How does the model's prediction quality and computation scale with significantly more particles, larger scenes, or more diverse material properties than those in training? Understanding this scaling behavior would clarify the practical significance of replacing kNN with space-filling curves.
- (Related to W4) For the real-world HO-Cap experiments, what are the quantitative metrics? Given that the encoder must be replaced entirely (MVSplat --> DepthAnything3), how much of the proposed pipeline is actually validated in the real-world setting versus only the dynamics model component?

**Limitations:**

yes

**Strengths And Weaknesses:**

### **Strengths**
- S1. (Originality) Replacing GNN-based kNN graph construction with PTv3's space-filling curve serialization is well-motivated, and the temporal merging mechanism that extends serialization to the spatiotemporal domain via bit-shifting of temporal codes is an attractive design for enabling cross-timestep attention without per-timestep transformer modules.
- S2. (Soundness) The end-to-end differentiable pipeline trained entirely with image reconstruction loss without ground-truth trajectories, depth, or physics priors is a strong experimental condition that demonstrates the framework's ability to discover physical structure from vision alone.
- S3. (Significance) The out-of-distribution generalization results are compelling: ground removal producing physically plausible freefall (Fig. 9), multi-object composition from single-object training (Fig. 10), and emergent shadow modeling (Fig. 11) provide evidence that the representation captures spatially grounded physics rather than appearance correlations.
- S4. (Presentation) The paper is generally well-written, and the positioning discussion carefully distinguishing 3DGSim from dynamic scene reconstruction and physics-prior methods is thorough and helpful.

---

### **Weaknesses**
**Major**
- W1. (Soundness) All three evaluation datasets are synthetic with relatively simple scenes: single objects on flat surfaces with uniform backgrounds (6 object categories each). The preliminary HO-Cap results (Section 5.4) are qualitative only, no quantitative metrics are reported. The gap between the controlled synthetic setting and realistic deployment scenarios (complex backgrounds, varying lighting, diverse materials) remains substantial and unaddressed quantitatively.
- W2. (Soundness) The evaluation metrics (PSNR, LPIPS, SSIM) measure rendered image quality, not physical accuracy. High PSNR does not guarantee physically correct trajectories. The model could produce visually plausible but physically incorrect dynamics. No trajectory-level evaluation (e.g., particle position error, energy conservation, momentum transfer) is provided. For a paper claiming to learn physical interactions, this is a notable gap.

**Moderate**
- W3. (Significance) The training cost is substantial (~6 days on a single H100 per dataset) for relatively simple scenes with only ~1000 short trajectories. The paper does not discuss scaling behavior with scene complexity or object diversity, and training appears dataset-specific with no evidence of cross-domain transfer.
- W4. (Soundness) The multi-view camera requirement (4 input views from 12 total cameras) is a significant practical limitation. Table S2 shows that reducing to 2 views out of 4 cameras results in training failure, and real-world synchronized multi-view capture is expensive and restrictive.

---

---

> ### Author Rebuttal · Authors · 2026-03-30
>
> ## W1: Real-World Evaluation and Dataset Limitations
>
> We agree with the reviewer that broader real-world analysis is an important next step. While we present quantitative results in E.5 in the appendix, the Ho-CAP dataset, being currently the only available multi-view real-world data set with human action conditioning, contains ~60 trajectories which is too small to support a quantitative in-depth analysis. We are currently working on creating larger real-world datasets for follow-up work. That said, despite the tiny dataset size, our model produces visually plausible simulations that are consistent across multiple views, which we believe to be a meaningful initial indication of real-world applicability of the proposed framework.
>
>
>
> ## W2: Beyond image based metrics, why not compare to GT state
>
> We thank the reviewer for raising this important point.
>
> This work aims to provide a scalable, image-only approach to learning simulators; without physics inductive biases, proprietary simulator access, hand-engineered features, or explicit object handling. A key consequence of this design is that our model **learns its own latent particle representation**, which does not have a one-to-one correspondence with the MPM simulator's internal state. The particles differ in number, spatial distribution, and semantic meaning. Therefore, computing particle-level position error against the simulator's ground-truth states is not possible as it would require establishing a bijective correspondence between two fundamentally different representations, which is ill-defined in our setting.
>
> This challenge is not unique to our work. Prior vision-based learned simulators; including VPD (Whitney et al., 2023) and HD-VPD (Whitney et al., 2024), all evaluate exclusively with image-based metrics (PSNR, SSIM, LPIPS) for the same reason. To our knowledge, no vision-based learned particle simulator has reported trajectory error against a reference solver.
> We also note that multi-step rollout image quality is itself an indirect measure of dynamics accuracy, as physically incorrect trajectories lead to compounding visual errors over longer horizons. To better capture temporal coherence beyond per-frame metrics, **we have additionally included FVD (Fréchet Video Distance)**, see W4 @ Reviewer n6s6, which evaluates the distributional quality of predicted video sequences.
>
>
>
> ## W3: Scalabilty and cross-domain transfer
>
> We thank the reviewer for this question. Regarding training cost, 6 days on a single H100 is lower than training costs reported for similar vision-based learned simulators (e.g., VPD, HD-VPD). We note that our framework replaces kNN graph construction; which accounts for 54% of forward time in GNN-based approaches (Wu et al., 2024b); with space-filling curve serialization, yielding near real-time inference (~16-20 FPS) on a single GPU.
>
> On scaling: although trained on single-object interactions, our combinatorial generalization experiments (Figs. 10, S6–S9) show the same model simulating up to 5 objects with plausible multi-body collisions, without any multi-object training data. This suggests that the **learned dynamics transfer compositionally to significantly larger scenes**. Prior work (Sanchez-Gonzalez et al., 2020) demonstrates cross-domain transfer for learned particle simulators; we expect similar transfer properties for our transformer-based architecture, though systematic cross-domain experiments remain an interesting direction for future work.
>
> Regarding material diversity: our model already handles rigid, elastic, and cloth dynamics. Scaling to more diverse material properties, and lighting conditions is an important next step, though to our knowledge no existing real-world multi-view dataset provides the material and lighting diversity needed to benchmark this systematically.
>
>
>
> ## W4: Camera requirements and the switch to DepthAnything3
>
> Considering the small amount of real-world examples in available multi-view data sets, training MVSplat with few cameras from scratch is not feasible. With too less data, depth prediction collapses to local optimums where 2D structures are placed in front of each camera. Though, as we show in the examples of the HOCAP dataset, the training also works for as few as 3 cameras.
>
> The quantitative metrics are presented in section E.5 in the supplementary. The training pipeline stays unchanged, only that the MVSplat was replaced with a pretrained DepthAnthing3 model which we finetune.

---

> > ### Author Rebuttal · Reviewer_t9ck · 2026-04-04
> >
> > I thank the authors for their thorough and well-reasoned rebuttal, which addresses all of my questions and concerns in a clear and satisfactory manner.
> >
> > - **W1 (Real-world evaluation):** The rebuttal acknowledges that broader real-world evaluation is an important next step and explains that the HO-CAP dataset is currently too small for quantitative in-depth analysis. While synthetic-only quantitative evaluation remains a limitation, the explanation is reasonable given the current availability of multi-view real-world datasets, and the visually plausible multi-view consistent results on HO-CAP provide a meaningful initial indication.
> >
> > - **W2 (Image-only metrics):** The rebuttal provides a well-justified explanation: since the model learns its own latent particle representation without one-to-one correspondence to the simulator's internal state, computing trajectory-level position error is fundamentally ill-defined. The observation that prior vision-based learned simulators (VPD, HD-VPD) also evaluate exclusively with image-based metrics for the same reason, and the addition of FVD as a temporal coherence metric, adequately contextualize this design choice. This remains an inherent limitation of the vision-only simulation paradigm, but is not a flaw of this specific work.
> >
> > - **W3 (Scalability):** The rebuttal clarifies that the training cost (6 days on a single H100) is lower than comparable methods (VPD, HD-VPD), and that the kNN replacement accounts for 54% of forward time savings in GNN-based approaches. The compositional generalization experiments (up to 5 objects from single-object training) provide encouraging evidence of scaling, though systematic cross-domain transfer experiments remain future work.
> >
> > - **W4 (Camera requirements):** The clarification that the pipeline works with as few as 3 cameras, and that the real-world adaptation preserves the full pipeline with only the encoder replaced by a finetuned DepthAnything3, addresses my concern about how much of the framework is validated in realistic settings.
> >
> > As my concerns have been fully resolved, I maintain my score of 4 (Weak Accept), recognizing the solid core contributions while noting the inherent evaluation scope limitations of the vision-only simulation paradigm.

---

### Decision · Program_Chairs · 2026-04-30

**Decision:**

Accept (regular)

**Comment:**

This paper received mixed scores of 3 weak accepts and 1 weak reject. After rebuttal, the remaining concerns are mainly about the experiments: missing comparisons with the per-scene optimization methods and lacking complex real-world demos. While these are issues that could be further improved, the key contributions remain valid despite of these experimental limitations, and also the authors provided comparison with Spring-Gaus in response to several requests. Therefore, the AC recommends acceptance.